# ALIGNING VIDEO MODELS WITH HUMAN SOCIAL JUDGMENTS VIA BEHAVIOR-GUIDED FINE-TUNING

## ABSTRACT

Humans intuitively perceive complex social signals in visual scenes, yet it remains unclear whether state-of-the-art AI models encode the same similarity structure. We study (Q1) whether modern vision and language models capture human-perceived similarity in social videos, and (Q2) how to instill this structure into models using human behavioral data. To address this, we introduce a new benchmark of over 49,000 odd-one-out similarity judgments on 250 three-second video clips of social interactions, and discover a modality gap: despite the task being visual, caption-based language embeddings align better with human similarity than any pretrained video model. We close this gap by fine-tuning different vision transformers on these human judgments with our novel hybrid triplet–RSA objective using low-rank adaptation (LoRA), aligning pairwise distances to human similarity. This fine-tuning protocol yields significantly improved alignment with human perceptions on held-out videos in terms of both explained variance and odd-one-out triplet accuracy. Variance partitioning shows that the fine-tuned video model increases shared variance with language embeddings and explains additional unique variance not captured by the language model. Finally, we test transfer via linear probes and find that human-similarity fine-tuning strengthens the encoding of social-affective attributes (intimacy, valence, dominance, communication) relative to the pretrained baseline. Overall, our findings highlight a gap in pretrained vision models' social recognition and demonstrate that behavior-guided fine-tuning shapes video representations toward human social perception.

## 1 INTRODUCTION

Humans effortlessly perceive the visual social world with remarkable nuance: we readily distinguish whether two people are comforting each other, collaborating, or competing, all by watching brief interactions. Humans can rapidly extract abstract information about intention, affect, and context, far beyond surface-level motion or pose information (Canessa et al., 2012; Lee Masson & Isik, 2021; McMahon et al., 2023). As AI systems increasingly interpret and interact in human-centered environments, aligning their representations with human social perception is essential. Yet, it remains unclear whether state-of-the-art models perceive social similarity the way humans do.

In this work, we investigate: **(Q1)** To what extent do current pretrained video and language models capture human-perceived similarity between social videos? **(Q2)** How can we instill a more human-like similarity structure into a video model using human behavioral data?

To address these, we introduce a new dataset of 49,484 human odd-one-out (OOO) triplet similarity judgments over 250 short (3s) videos depicting everyday social scenes. Each triplet judgment identifies which of three videos is least like the others, inducing a behavioral similarity structure over the video set. Remarkably, we find that embeddings from a language model applied to video captions outperform all pretrained video model embeddings at predicting these judgments, despite the human task being presented in a purely visual manner. To close this gap, we then propose a behavior-guided fine-tuning strategy that incorporates human similarity judgments directly into video model training. We introduce a hybrid loss combining: (i) Triplet loss, enforcing local alignment with human triplet OOO comparisons; (ii) representational similarity analysis (RSA) loss, aligning the global pairwise embedding structure with human representational similarity matrices (RSMs). Using parameter-efficient low rank adaptation (LoRA) (Hu et al., 2022), we fine-tune a TimeSformer video model

with $< 2$ parameter updates. Our approach substantially improves human-model alignment: fine-tuning explained variance increases by 42.67% relative to the pretrained baseline (on average, see Appendix §F), approaching the behavioral reliability ceiling, and surpasses language model performance. Variance partitioning shows that a fine-tuned video model more strongly overlaps with the language model, compared to the pre-trained baseline, and explains additional variance in human judgments not captured by the language model.

**Contributions.** In this work, we make three main contributions: (1) We introduce a benchmark of ∼49k human odd-one-out judgments on social video clips, providing the first large-scale dataset of human-perceived similarity in videos. (2) We propose a geometry-level training method that combines triplet supervision with a differentiable RSA objective to directly shape video representation spaces, and is applicable to a range of vision transformers. (3) We provide empirical evidence that behavior-guided fine-tuning achieves near-ceiling alignment with human similarity judgments, surpassing the best language model.

## 2 RELATED WORK

**Human Similarity Judgments in Vision.** Measuring how humans perceive similarity among stimuli has long been a tool to probe mental representations (Biederman, 1987; Edelman, 1998; Nosofsky, 1986; Goldstone, 1994; Hebart et al., 2020). Large-scale behavioral studies have mapped out the "similarity space" humans use for objects and scenes. Prior work has used odd-one-out (OOO) and triplet tasks to reveal the latent structure of human perception in domains such as objects (Hebart et al., 2020), "reachspaces" (reachable interaction environments; Josephs et al., 2023), and materials (Schmidt et al., 2025). The majority of prior work focuses on the similarity structure of static image content. Our work extends this approach to social video. One prior study has investigated human judgments of dynamic stimuli and found that these judgments rely more on social-affective features than surface visual or scene features (Dima et al., 2022). While this prior work has modeled dynamic similarity judgments it has focused on explaining human judgments rather than model alignment.

**Aligning Models with Human Perception.** There is growing interest in aligning model representations with human cognitive representations, with the goal of improving interpretability and performance. Recent work has also highlighted that optimization on engineering tasks does not necessarily improve model alignment (Garcia et al., 2025; Linsley et al., 2023). Most efforts at human-alignment rely on direct human feedback, for example reinforcement learning from human feedback for generative video or text-to-video models (Kaufmann et al., 2023; Liu et al., 2025a). Such supervision optimizes task rewards or output quality, but does not necessarily constrain the internal geometry of representations. These approaches are often data-intensive/require a human in the loop (Furuta et al., 2024; Li et al., 2024).

Odd-one-out similarity judgments, in contrast, provide richer relational supervision: each triplet encodes a relative comparison that reflects latent social structure, rather than scalar preferences alone. Muttenthaler et al. (2023) show that globally aligning model similarity to human judgments yields more interpretable features, but focus on static images. Further, a recent model, DreamSim (Fu et al., 2023), learns perceptual similarity from synthetic image pairs. Through finetuning an embedding space to these human judgments produced a metric that both (1) aligned better with human perception and (2) improved overall image retrieval performance. These methods highlight the value of human-derived similarity signals, but they largely remain limited. So, instead of focusing on static images, low-level perceptual comparisons, or synthetic domains, our work targets *dynamic, naturalistic, social videos*. This allows the models to learn similarity structure *directly* from human judgments of social interactions.

More recently, the focus of alignment across multimodal settings has moved toward large-scale preference learning and cross-modal supervision in vision-language models. Contemporary advances include adaptive vision-enhanced preference optimization (Lu et al., 2025), retrieval-augmented direct preference optimization (Xing et al., 2025), online preference generation for failure-driven negative sampling(Liu et al., 2025b), peer-based preference evaluation using a panel-of-models (Hernandez et al., 2025), and token-level inference-time alignment guided by learned reward models (Chen et al., 2025). These approaches largely align models to *task preferences, instruction-following behavior,*

*or general response quality.* But, they often do not pay particular attention to the relational structure that underlies human judgments, especially when it comes to social interactions. Hence, our work is complementary to these developments: rather than aligning free-form outputs or task responses, we instead focus on shaping video representations so that distances in embedding space reflect the human similarity structure of naturalistic social behavior.

**Beyond categorical video pretraining.** Recent work has focused on models containing transformer based architectures and large scale pretraining (TimeSformer; Bertasius et al., 2021), (ViViT; Arnab et al., 2021), and (VideoMAE; Tong et al., 2022). Although they achieve state-of-the-art results on action classification benchmarks, their training objectives underscore category-level recognition (e.g., "dancing" vs. "cooking") instead of higher-level aspects of social behavior (e.g., intentions, affect, or interaction dynamics). On the other hand, recent multimodal video-language models such as VideCLIP (Xu et al., 2021) and All-in-One (Wang et al., 2022), feed textual supervision to video embeddings, giving way to semantic abstractions that are not easily derivable from raw video. These video-language approaches still depend on captions and descriptions, which do not always track the relational or affective signals people rely on when comparing social scenes. Self-supervised methods like V-JEPA (Assran et al., 2025) take a different route by predicting upcoming content, pushing the model toward representations that carry temporal and semantic detail without relying on text. Other research has expanded the scale of video-language pairing (Rizve et al., 2024), used caption perturbations to increase robustness (Bansal et al., 2023), or introduced human preferences to guide generative models (Wang et al., 2024). Still, none of these efforts tune representations according to how people compare and group natural social interactions.

## 3 METHOD

Our approach has two stages: (1) Measure human-perceived similarity, where we collect odd-one-out judgments on video triplets to construct a human similarity matrix; and (2) Leverage behavior-guided fine-tuning on a video model with the objective of matching its embedding distances to human similarity structure. This is achieved through a hybrid loss fucntion that enforces local triplet constraints within global alignments of similarity matrices. (Fig. 1).

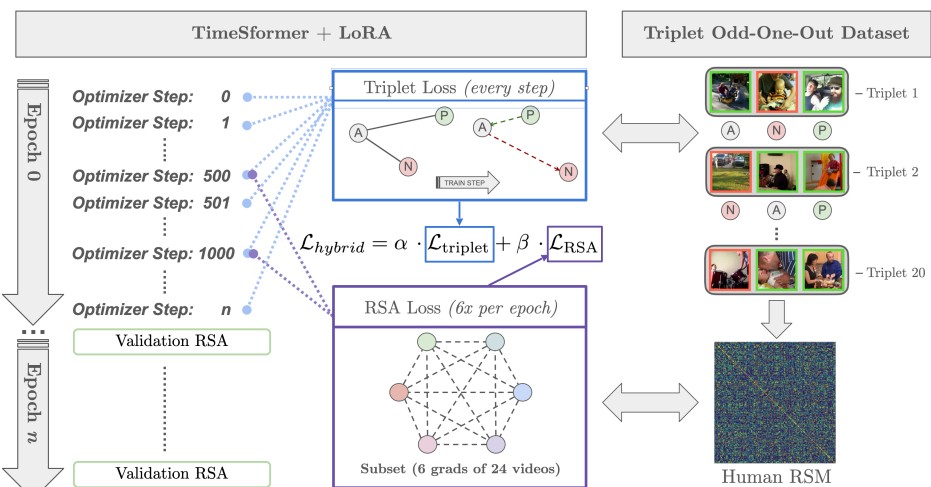

Figure 1: Triplet Odd-One-Out Dataset and TimeSformer Hybrid fine-tuning. Similarity judgments are derived via a triplet odd-one-out task where human choices are used as a positive or negative signal for each training loss. Moreover, the model is updated with a triplet loss (blue) on a batch of Anchor (A), Positive (P), Negative (N) triplets. Six times per epoch, we apply an additional RSA loss (purple) on a subset of 24 videos, 6 of which as gradients by aligning the model's pairwise distances with the human similarity derived from all triplets. The combined training objective of triplet and RSA loss is defined in Eq. 5.

## 3.1 Human Similarity Judgment Dataset

We introduce a novel, large-scale dataset of human similarity judgments of short video clips. The stimulus set consists of 250 short video clips (3 seconds each) depicting a wide range of everyday human activities and social situations from a publicly available dataset (McMahon et al., 2023; Garcia et al., 2025), a subset of the Moments in Time dataset (Monfort et al., 2019), densely labeled with human social judgments. Each video was paired with a brief descriptive caption (one sentence summarizing the action) to evaluate language models (see Appendix §G.1).

We use a triplet odd-one-out paradigm to gather similarity judgments (Hebart et al., 2020). In each trial, a participant saw three videos (see Appendix B), and were asked to "focus on what the people are doing and choose the odd-one-out". By choosing the odd-one-out, the participant implicitly indicated that the other two were more similar to each other. This triplet-based method yields more information per trial than a simple pairwise rating. 245 human participants were recruited online via our University's psychological research platform, and participated in the study on Meadows Research (`https://meadows-research.com`). All participants gave informed consent in accordance with our internal Institutional Review Board, who provided explicit approval of all protocols and procedures discussed. All participants were 18 or older, with normal or corrected-to-normal vision, and at least academically proficient English speakers (see Appendix Fig. 6).

For model training and evaluation, judgments were split based on the pre-determined stimulus split released with the benchmark: 200 train videos (24,096 triplets) and 50 test videos (368 triplets). For both train and test set of judgments, we calculated a $200 \times 200$ human similarity matrix $\mathbf{S}^{(human)}$ and a $50 \times 50$ human similarity matrix, respectively. Following prior work (Hebart et al., 2020), we define the human similarity between two videos as the probability (or frequency) that they were judged together (not odd-one-out) in triplet trials.

**Choice of Distance Metric.** Because embeddings from different architectures vary widely in scale and norm, we use cosine similarity as the primary pairwise metric. For a video $v$ with embedding $f(v)$, the similarity between videos $i$ and $j$ is:

$$S_{ij}^{\text{(model)}} = \cos\big(f(v_i),\ f(v_j)\big). \tag{1}$$

Cosine similarity emphasizes the angular relationship between vectors, effectively normalizing differences in magnitude across features. This is particularly useful when comparing across layers or across different architectures, where feature norms may differ systematically. Empirically, we found that cosine similarity correlates more strongly with human judgments than Euclidean distance, in line with prior work on representational alignment (Hebart et al., 2020; Kriegeskorte et al., 2008).

## 3.2 Representations from Video and Language Models

We evaluate pretrained models on how well their layer-wise embeddings aligned with the human similarity structure (Q1). For each model layer, we obtained a feature embedding for each video (or sentence caption) and computed an analogous $50 \times 50$ similarity (or distance) matrix, for comparison to the human test set RSM with RSA (Kriegeskorte et al., 2008).

We evaluate 9 pretrained vision models including both CNN-based and Transformer-based video encoders. For example, X3D-M – a CNN from the X3D family optimized for efficient video classification (Feichtenhofer, 2020), SlowFast – a two-pathway CNN capturing both slow and fast temporal dynamics (Feichtenhofer et al., 2018); and TimeSformer – a video Transformer that factorizes spatial and temporal attention trained on Kinetics-400 (Kay et al., 2017; Bertasius et al., 2021). We feed each 3s clip into these models (after resizing frames to the required model resolution). We take the model's embeddings at every layer, utilizing the DeepJuice software package (Conwell et al., 2024) for efficient layer-wise calculations. For fairness, we ensure each embedding is a vector of comparable dimension by down-sampling using sparse random projection (SRP) based on the Johnson–Lindenstrauss (JL) lemma with $\varepsilon = 0.1$. This automatically sets the projection size according to the number of samples, yielding 4,732 dimensions for the training split ($N = 200$) and 3,353 dimensions for the test split ($N = 50$), which preserves pairwise distances within $\pm 10\%$ with high probability. To select the evaluation layer, we perform a 5-fold cross-validation on the 200-video training set, choose the layer with the highest mean Spearman's $\rho$ across folds, and then fix that layer for evaluation on the held-out 50-video test set.

For each video, we also obtain a representation from a language model based on the video's caption. We selected 22 widely used transformer-based language models spanning sentence vs. retrieval objectives, parameter scales, and multilingual coverage, yielding a diverse and reproducible set of off-the-shelf caption encoders for comparison. (see Appendix Tab. 2) and similarly compute a similarity matrix for the captions based on cosine similarity between the layer-wise embeddings. We include the top language model's (`paraphrase-multilingual-mpnet-base-v2`) alignment performance as a point of comparison to video models (Appendix Tab. 2).

In addition we selected two modern image models to benchmark, CLIP (vision transformer only) and Dino-ViT. We extract seven equally spaced frames across the video and average across the embeddings before continuing with the same evaluation pipeline as video and language.

We also selected one vision-language model, Qwen3-VL-2B-Instruct to compare as a SOTA multimodal point of reference. We selected this specific parameter count as it is the closest match to TimesFormer. We input both the video caption and video frames (using the procedure described for image models agove) to the model and extract both embeddings for RSA.

### 3.3 Behavior-Guided Fine-Tuning of the Video Model

Our core approach for (Q2) is to fine-tune vision models using the human judgments as supervision. We focus on the transformer architecture with the highest pretrained performance (TimeSformer). We apply a lightweight fine-tuning strategy with LoRA, updating less than 2% of the model's parameters (1.9M trainable vs. 123M total) while keeping the other 121M parameters frozen. This approach inserts low-rank matrices into each attention layer (rank = 16), enabling efficient adaptation with minimal compute overhead and reduced risk of overfitting to our dataset. We also report fine-tuning results for the highest performing image model (CLIP), and a more recent video transformer in VideoMAE (Tong et al., 2022).

#### 3.3.1 Hybrid Loss Function

We design a loss $\mathcal{L}_{\text{hybrid}}$ that combines a triplet loss term ($\mathcal{L}_{\text{triplet}}$) and an RSA loss term ($\mathcal{L}_{\text{RSA}}$) to address both local and global alignment (Fig. 1).

**Shared notation and distance.** Let $f(v)$ be the embedding of video $v$. We use $\ell_2$-normalized embeddings $\mathbf{z}_i = f(v_i)/\|f(v_i)\|_2$ and define a single cosine-distance operator that is shared by both losses:

$$d(i,j) = 1 - \langle \mathbf{z}_i, \mathbf{z}_j \rangle. \tag{2}$$

**Triplet Loss (local constraints)** For each human odd-one-out judgment we seek to minimize the distance between anchor video $i$ and its positive pair $j$ to be less than the distance to its negative pair $k$ (odd-one-out) by a margin of $\gamma$. Specifically, we penalize violations of a margin $\gamma = 0.2$:

$$\mathcal{L}_{\text{triplet}}(i,j,k) = \max\big\{0,\, d(i,j) - d(i,k) + \gamma\big\}. \tag{3}$$

**RSA Loss (global geometry)** To shape the broader geometry toward human similarity structure, we inject an RSA step six times per epoch. At each RSA step, we sample a batch of $K{=}24$ videos $\mathcal{K}$ and designate a subset of $M{=}6$ indices $\mathcal{G} \subset \mathcal{K}$ whose embeddings carry gradients. We limit gradients to $M{=}6$ to keep memory and runtime manageable while still providing ample supervision: each RSA step considers all pairs that include one of these six videos (up to 123 pairs before masking), which we found gives a strong signal without the overhead of updating all 24 items.

We calculate model RDM entries with $d(\cdot,\cdot)$ for all unordered pairs $\{i,j\} \subset \mathcal{K}$ with $i \neq j$ and $i \in \mathcal{G}$ or $j \in \mathcal{G}$. Corresponding human distances $d^{\text{H}}(i,j)$ are taken from the split-specific behavior RDM, masking out pairs without judgments to create a masked index set $\mathcal{M}$.

The RSA loss is the negative RSA score between the $z$-scored model and human distances of the masked index set $\mathcal{M}$:

$$\mathcal{L}_{\text{RSA}} = -\text{corr}\Big(z\big(\text{vec}(d)\big)[\mathcal{M}],\, z\big(\text{vec}(d^{\text{H}})\big)[\mathcal{M}]\Big), \tag{4}$$

where $\mathrm{vec}(\cdot)$ denotes vectorization of the upper triangle, and $z(\cdot)$ denotes per-step standardization to zero mean and unit variance.

Pearson correlation is used for the RSA loss during training to ensure the loss is differentiable.

**Hybrid Loss.**    We combine the triplet (local) and RSA (more global) supervision with a weighted objective:

$$\mathcal{L}_{\text{hybrid}}^{(t)} \;=\; \alpha\,\mathcal{L}_{\text{triplet}}{}^{(t)} \;+\; \mathbb{1}_{\text{RSA}}(t)\,\beta\,\mathcal{L}_{\text{RSA}}{}^{(t)}, \tag{5}$$

where $\mathcal{L}_{\text{triplet}}$ captures fine-grained constraints from odd-one-out judgments and $\mathcal{L}_{\text{RSA}}$ encourages broader geometric alignment on sampled subsets. The indicator $\mathbb{1}_{\text{RSA}}(t)$ equals 1 if step $t$ is one of the scheduled RSA steps and 0 otherwise. Specifically, we compute the total number of optimizer steps in an epoch, divide by 6, and activate the RSA loss at these evenly spaced intervals. We fix $\alpha = 0.7$ and linearly ramp $\beta$ from 0.3 to 0.7 over training epochs. We do this to emphasize the local triplet loss early and ensure the model starts by getting the odd-one-out relationships correct, then gradually increase the weight of the global RSA loss ($\beta$) as training progresses to fine-tune the overall similarity structure.

**Training Procedure.**    We fine-tune for 50 epochs with AdamW (see Loshchilov & Hutter, 2017) with learning rate $= 1 \times 10^{-4}$, mixed precision, and gradient-checkpointing, using a batch size of 4. At each optimizer step, we apply the triplet loss; the RSA term is injected periodically as described above. We select the best checkpoint by RSA validation performance on a held-out 20% split of the training judgments (monitoring explained variance $R^2$). For ablations, we also train models with triplet-only and RSA-only objectives under the same optimizer and schedule.

### 3.3.2    OUT-OF-DISTRIBUTION LINEAR PROBES FOR SOCIAL-AFFECTIVE ATTRIBUTES

To see if human similarity alignment improves the model's human alignment with other, out-of-distribution, tasks, we use human annotations for five key attributes of social scenarios included in the video dataset (McMahon et al., 2023): *Intimacy* (how intimate/personal the interaction is), *Valence* (overall emotional positivity vs negativity), *Arousal* (energy or intensity of the action), *Dominance* (power dynamic between people), and *Communication* (whether people in the video are communicating with one another). Multiple annotators independently rated, averaged, and $z$-scored each of the 250 videos on these scales. We use a ridge regression linear probe on layer-wise model embeddings with the same train-test split for the models as main experiments.

### 3.3.3    ACTION-RECOGNITION EVALUATION

To ensure human-aligned fine-tuning does not lead to catastrophic forgetting on the original task, we evaluate the baseline and fine-tuned video models' action recogition performance, following the UCF101 benchmark (Soomro et al., 2012) split1 (101 action categories). We freeze the model backbones (both pretrained and fine-tuned with LORA adapters), extract model embeddings, and train a linear probe on UCF101 split1 across three seeds (Top-1 accuracy; mean±sd, see Appendix E).

## 4    RESULTS

### Q1: DO PRETRAINED MODELS CAPTURE HUMAN-PERCEIVED SIMILARITY?

On average, both language and video models show a modest ability to capture human video similarity judgments. Even state of the art (SOTA) vision-language models like Qwen3 modestly capture similarity judgments, but still fall shorter than Language only. (Fig. 2). Among pretrained baselines, the best caption–based language embedding (*paraphrase-multilingual-mpnet-base-v2*) achieves higher explained variance ($R^2 = 0.134$) *and* higher OOO accuracy (70.38%) than the best pretrained video model (TimeSformer: $R^2 = 0.102$; OOO $= 63.59\%$; Appendix Tab. 2). Thus, even though human participants performed a purely visual task without captions, their judgments were better predicted by text embeddings, suggesting critical gaps in pretrained video models.

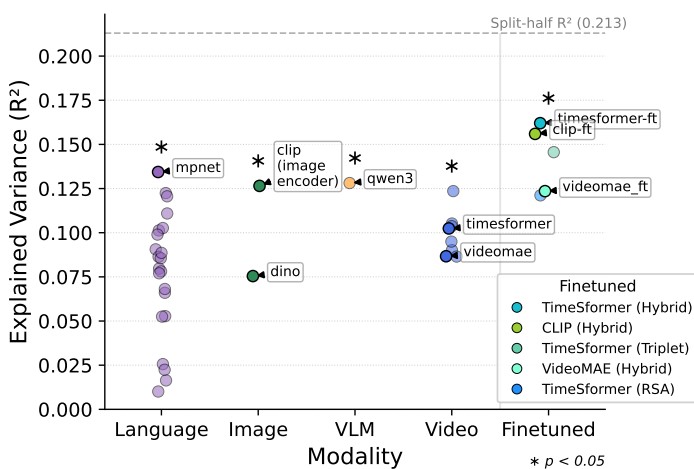

Figure 2: Explained variance ($R^2$) computed as Spearman's rank correlation between model embeddings and human similarity judgments and we report its square as a measure of explained variance (differing from regression). Horizontal dashed line shows the split-half spearman correlation[2] of the human RSM used as our noise ceiling (see §C.3).

### Q2: HOW CAN VIDEO MODELS LEARN HUMAN-LIKE SIMILARITY?

We next ask whether we can imbue video models with more human-like similarity structure via fine-tuning. To use the LORA procedure, we select TimeSformer as the best performing transformer model (see §3: Methods). Fine-tuning with hybrid triplet-RSA loss shows a significant improvement over the pretrained TimeSformer baseline in terms of both correlation and accuracy. Importantly, the hybrid fine-tuned video model outperforms all pre-trained models, including the best *language-based caption embeddings* both in terms of $R^2$ and OOO accuracy (Fig. 2; Appendix Tab. 2).

The hybrid loss also outperforms both the triplet-only and RSA-only fine-tuning, showing that the combination of local and global constraints is more effective than either alone (Fig. 2). Importantly, the triplet-budget-matched control achieved performance better than triplet-only but below hybrid, demonstrating that RSA contributes more than simply additional training signal.

In addition to TimeSformer we selected two other models to perform the same hybrid finetuning method on. VideoMAE, another video model, and the best performing Image model from the benchmark, CLIP-ViT-b16. Both models improved human alignment significantly. VideoMAE increased from 0.08 to 0.12, while CLIP increased from 0.12 to 0.15, making it the second best model after TimeSformer. This shows that our hybrid finetuning method can scale beyond only TimeSformer and provide valuable gains among Image and Video models.

Moreover, we perform a variance partitioning analysis using the best language model's embedding as a reference (Fig. 3). We fit a multiple regression predicting human similarity distances for all video pairs using model distances as predictors. By comparing explained variance ($R^2$) across regression models, we es-

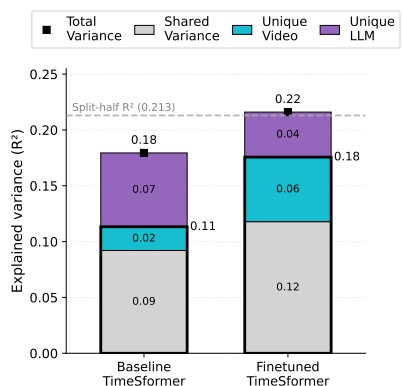

Figure 3: Variance partitioning before and after finetuning. Finetuning increases unique TimeSformer variance (cyan), reduces unique language model variance (purple), and expands shared variance (gray). Total explained variance (black markers) approaches the reliability ceiling. Black outline indicates total variance explained by the video model.

timated unique and shared contributions of the pretrained TimeSformer, the fine-tuned TimeSformer, and the language model. In the **pretrained (baseline) case** (left), the video model contributes little unique variance, with most of its explanatory power overlapping with the language model and the language model still accounting for substantial unique variance on its own. In the **fine-tuned case** (right), shared variance between models increases and the video model captures more unique variance (see Appendix Tab. 3). These results suggest that fine-tuning both aligns the video model more closely with language-derived semantic structure and enables it to encode additional social–visual nuances that are less easily captured by caption embeddings.

**Encoding of Social-Affective Attributes.** To test whether fine-tuning enhances the encoding of social and emotional factors of the videos, we run linear probes predicting five attributes often emphasized in human descriptions of social interaction: intimacy, valence, arousal, dominance, and communication.

As shown in Fig. 4, fine-tuning substantially improves the model's sensitivity to social-affective dimensions. The largest gains appear in *Valence* and *Dominance*, while *Intimacy* was already well-encoded even before fine-tuning. *Communicating* shows modest improvement, whereas *Arousal* remains relatively unchanged. Notably, the model was never trained on these human judgments. Its improvement therefore suggests that human similarity judgments were themselves shaped by these underlying factors, and highlights how similarity-based supervision encourages the emergence of interpretable, socially meaningful features.

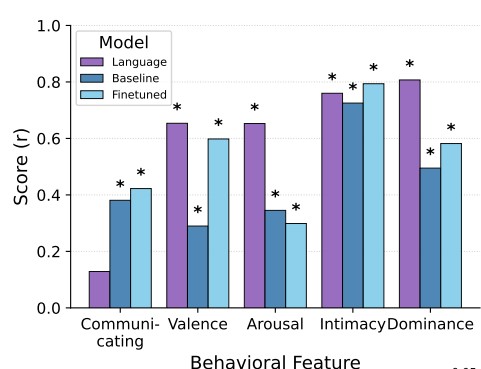

**Action recognition performance** The pretrained TimeSformer achieved $95.75 \pm 0.18\%$ Top-1 accuracy with a frozen linear probe across three seeds, and the fine-tuned model achieved $95.70 \pm 0.14\%$ (on UCF101). The negligible difference (paired mean $\Delta = -0.05$ pp) confirms that behavior-guided fine-tuning preserves action recognition ability, with no catastrophic forgetting.

Figure 4: Pearson correlation ($r$) scores for predicting social-affective attributes from video embeddings using Ridge Regression. Language (purple) is the best performing language model for comparison to baseline (dark blue) and finetuned (light blue) TimeSformer.

## 5 DISCUSSION

Our findings reveal a substantial mismatch between how current video models and humans perceive social video clips, and demonstrate a practical route to reduce this gap via behavior-guided fine-tuning. We created a new dataset of human video similarity judgments and presented an approach to align video model representations with humans. We found that while pretrained video models already capture some aspects of human similarity, they lag behind language-based embeddings. To close this gap, we fine-tuned a video transformer using a combination of triplet and RSA losses derived from human judgments, resulting in a model that more closely reflects human notions of similarity. This fine-tuned model not only aligns better with human judgments in aggregate, but also generalizes to better match judgments of high-level social-affective concepts, as evidenced by linear probe analyses. Variance partitioning further revealed that fine-tuning shifted the video model toward the semantic structure captured by language model embeddings while also contributing unique explanatory variance not captured by language models, indicating a unique contribution of visual information to this task.

## 5.1 Human Alignment as Supervision

Our approach frames human similarity judgments as a distinct form of supervision: instead of predicting explicit labels, the model is guided to organize its representation space to mirror human relational structure. This complements categorical labels by encouraging the geometry to capture factors humans intuitively use, such as social or affective context. Compared to alternatives like attribute annotation (e.g., intimacy or scenario type), this method is holistic: humans integrate multiple cues when judging similarity, and alignment recovers that integrated structure without enumerating each factor. Our social probe experiments also showed the fine-tuned model learned attributes it was never directly trained on. Interestingly, prior work has shown that video models struggle to match these attributes (Garcia et al., 2025), highlighting a particular benefit of fine-tuning for improving social judgments. Similar benefits from human similarity supervision have been demonstrated in prior work (Muttenthaler et al., 2023; Fu et al., 2023); our study extends these findings to social videos, areas that AI vision typically struggles with (Garcia et al., 2025).

## 5.2 Why Language Models Outperformed Video Models

Understanding social interactions often requires abstract inferences (goals, roles, affect) that go beyond visible motion. Video models, trained mainly for action classification, may emphasize kinematics and object cues, while caption-based language embeddings encode high-level semantics (e.g., "friends boxing for fun" vs. "strangers fighting angrily"). Humans likely rely on similar latent variables, which explains why language embeddings aligned more closely with human judgments. However, the fact that these are learnable by a video model, and that a fine-tuned video model can learn to explain human variance not attributable to language models, supports the idea that humans encode many aspects of this social structure visually (McMahon & Isik, 2023). When we compared the on the odd-one-out task of the same set of triplets on pre-trained vision and language models, the language model agreed with the human participants, while the vision model disagreed. After finetuning the vision model agreed with the human choice. For example, the video model tends to prioritize visual features, whereas humans and language models prioritize information about social relationships (Appendix § H). An open question is whether self-supervised video models trained via predictive representation learning may close this gap: recent work such as V-JEPA 2 (Assran et al., 2025) suggests promising progress in this direction. Comparing more modern video models to this dynamic human benchmark is a promising area for future video model evaluation.

**On the Hybrid Loss.** Our fine-tuning objective combines a triplet loss with an RSA loss, balancing local and global alignment. The triplet component ensures that fine-grained distinctions from the original model are preserved while pulling together pairs judged similar by humans. With the addition of the RSA component, it is complimented by aligning the model's overall pairwise structure with human RSMs. This distills the relational knowledge at a global level, reflecting the findings by Muttenthaler et al. (2023), who showed that constraints on global geometry that match human similarity yields better, and more interpretable task-effective features when preserving local structure. Where RSA is typically used as a tool for analysis (Kriegeskorte et al., 2008), our contribution takes it a step further and re-purposes it as a differentiable objective. With this, the hybrid loss leverages both local and global supervision to push the representation towards richer semantic space that is reflected by human judgments.

## 5.3 Limitations

**Dataset coverage.** Although diverse, the 250 videos in our dataset stem from a single source corpus. As such, claims to stronger robustness require testing to be transferred to other video datasets and even domains (especially those with varying styles, contexts and cultural settings). Cross-dataset evaluation will be important for assessing the generalization and broader applicability of human-aligned representations. The high prediction accuracy of our fine-tuned model suggests it may be used as a tool to generate synthetic similarity data on larger scale video datasets.

**Evaluator subjectivity.** Inherent differences in cultural background, personal experience, and attentional focus vary across individuals when conducting social similarity judgments. Therefore, our current model smooths over such heterogeneity by only capturing the aggregate consensus. While this may be useful for deriving stable group-level metrics, there will be limits at the individual level.

In future work, exploration of personalized alignment may prove useful and can be done by collecting repeated judgments from single users or by way of clustered annotations with similar perceptual styles, enabling models to reflect user or subgroup specific social perception.

**Task scope.** Our evaluation primarily focuses on similarity alignment along with a few attribute probes. The possibility of trade offs exist despite preliminary checks displaying competency in basic action recognition in the fine-tuned model. In principle, enhancing human alignment could reduce discriminative power on conventional benchmarks. Consequently, more comprehensive evaluation across multiple tasks and domains would be required. A principled safeguard such as multi-objective training (i.e., combining classification loss with alignment losses) would ensure retention of conventional task performance in models while gaining alignment with human similarity structures.

**Cultural specificity.** One limitation is that this work reflects a predominant Western context across both the stimuli from the Moments in Time dataset and the human judgments collected. All video annotations were provided by native English speakers recruited from Prolific, and all participants in the triplet odd-one-out experiment were recruited through a U.S. university's research platform, meaning they were either native English speakers or at least academically proficient English speakers. It is true that differences in social norms, interaction styles, and expectations about social behavior can lead to a great degree of variance in social similarity judgments (e.g., see Pang et al., 2024). So, the similarity structure learned by our models should not be assumed to generalize across all populations, whose views might differ based on their socio-cultural backgrounds. Future work should investigate how cultural priors shape interpretations of social scenes and interactions, and more importantly, the adaptability of these models to culturally specific or mixed similarity structures.

### 5.4 BROADER IMPACT

Video models that are aligned with human social similarity judgments provide a path to more trustworthy and intuitive AI systems. Embeddings tat align with human behavior may improve interpretability, video retrieval, and recommendation by way of organizing content that is reflective of human categorization. Our findings suggest that such alignment also promotes emergent encoding of social-affective features, with potential applications in affective computing and safety-sensitive domains. However, models that reflect human perception may also inherit human biases. While our dataset is diverse, culturally specific notions of similarity may also be encoded as a result of the aforementioned factors. This validates the deployment of broader studies that include analysis for bias with more diverse annotation sources, ensuring fairness and robustness across populations.

## 6 CONCLUSION

We present an approach to align vision transformer representations with human social perception by leveraging a new dataset of human similarity judgments. We find that pre-trained image and video models do not fully capture the nuanced similarity structure that humans perceive in social and action-centric videos, whereas language-based representations fare substantially better (Appendix Fig. 7). To close this gap, we fine-tune different vision transformers using a novel combination of triplet and RSA losses derived from human judgments, resulting in models that much more closely reflects human notions of similarity. We find that a fine-tuned video model not only aligns better with human judgments in aggregate (boosting correlation and odd-one-out accuracy, Appendix Fig. 8), but also internally encodes high-level social-affective concepts more clearly (as evidenced by linear probe analyses) and even captures new variance beyond what language-based features explain (as shown by our variance partitioning analysis). Our work demonstrates that incorporating human similarity data is a viable path to enriching model representations beyond what traditional supervised tasks achieve.

### REPRODUCIBILITY STATEMENT

The data we have collected on the odd-one-out similarity judgments (with the canonical 200/50 train-test split), along with human RSMs and video annotations on all 250 videos will be publicly released. The pertinent metrics for all of the models we have evaluated (both pretrained/baseline and finetuned) are provided in detail in §3, with additional details on RSA and variance partitioning in the

Appendix (§ C.2). Details on the availability for coding material concerning embedding extraction, similarity computation, and model evaluation (both for $R^2$ scores and odd-one-out accuracy) are provided in the Appendix (§ D). This section also outlines the way by which others could obtain the Moments in Time dataset we used for our analyses. After gaining access, the mapping we used between raw video files and their representative IDs (0-249) will also be available in the code base. Furthermore, we will also release the configuration details for our vision models (TimeSformer, CLIP, and VideoMAE) with LoRA adapters, including the scripts for hybrid, triplet-only, RSA-only, and triplet-budget-matched versions. See § 3 for a more high-level description of training details. Next, the validation and reporting metrics follow the procedure outlined in § 3.3.3-§ 4. To facilitate this, we will also release the scripts necessary for full pre-processing and training. You can find more information about the UCF101 linear-probe action recognition and social-affective probing experiments in Appendix § E). Finally, we wish to support both full retraining and more lightweight reproduction for accessibility. Therefore, we will also make available all training and evaluation code, as well as pretrained adapters and precomputed RSMs to reproduce these analyses.

## ETHICS STATEMENT

All procedures pertaining to the analyses conducted throughout this paper adhere to ethical standards. The behavioral data used was collected under the internal Institutional Review Board (IRB) approval. Informed consent was obtained from all subjects before their participation. The experiment itself was straightforward: they were instructed to make quick "odd-one-out" choices on 3-second video clips that showed everyday social interactions without any identifying details. We did not collect any personal data beyond demographic information, all of which were optional. We compensated the participants for their time appropriately, and made sure that their responses were reliable without putting too much strain on them. The data we obtained was not used to infer private characteristics on the participants, only for model-human representational alignment. We commit to releasing the dataset (except the actual videos due to licensing, see Appendix § D) and the code to encourage transparency and replication.

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

## A    LLM Usage

LLMs were used for minor edits, such as grammar, phrasing, and shortening, throughout this paper. LLMs were not used during the data collection or the analysis stages. No major task, such as ideation, full content generation, or substantive interpretation of results, was delegated to LLMs. All conceptual framing, experimental design, and analytical decisions were carried out by the authors.

## B    Triplet Selection Algorithm

$\mathbf{S}^{(human)}$ is an estimate (aggregate), rather than a fully observed matrix. This is because the triplet sample is sparse relative to all pairs $\binom{250}{2} = 31,125$. We came up with a specialized algorithm to create the triplets to ensure adequate coverage with the least amount of participants possible. So, we designed this procedure so that *every possible pair $(i, \ j)$ appears in at least one triplet $(i, \ j, \ k)$*. Since the similarity (probability) matrix is constructed through how many times a pair of videos was rated similar based on how many times they appeared, this guarantees that each "pair" would have at least one rating.

This problem is conceptually equivalent to a *set cover*. Namely, the universe of elements consists of all video pairs, and each triplet corresponds to a subset that covers three of those video pairs. Finding the truly minimal set of triplets that covers all possible pairs is NP-hard. So, we implemented a greedy approximation strategy to iteratively choose the most informative triplet at each step, given all the triplets selected before and remaining.

- First, we randomly sample a candidate pool for the triplets at each iteration.

- Then, from this pool, we select the triplet that covers the largest number of pairs (max: 3) *not yet included*.

- Next, we mark those pairs as "covered" and continue the iteration until every pair has been assigned at least one triplet.

We prioritized efficiency with this "greedy" search, since we effectively minimized the number of triplets (and thus the number of participants) we needed to guarantee full pairwise coverage. After coverage was achieved, we adjusted the total number of triplets by adding more so that it was divisible by 220 (11 sets of 20 trials for each participant).

$$\text{Minimum possible triplets} = \frac{\binom{250}{2}}{3} = \frac{31{,}125}{3} = 10{,}375.$$

Our greedy algorithm produced 10,780 triplets $\rightarrow \dfrac{10{,}780}{220} = 49$ participants required.

---

**Algorithm 1** Triplet Selection Covering All Pairs (Greedy Set Cover Approximation)

---

**Require:** Number of items $N$ (e.g., $N = 250$ for 250 video stimuli)
**Ensure:** Set of triplets $T$ covering all pairs, with $|T|$ divisible by 220
 1: $P \leftarrow \{(i,j) \mid 0 \le i < j < N\}$            ▷ All pairs
 2: $S \leftarrow \{(i,j,k) \mid 0 \le i < j < k < N\}$       ▷ All triplets
 3: $T \leftarrow \emptyset$                    ▷ Selected triplets
 4: **while** $P \ne \emptyset$ **do**
 5:     $C \leftarrow$ random sample of $\min(|S|, 10,000)$ triplets from $S$
 6:     $best\_triplet \leftarrow$ triplet in $C$ maximizing coverage w.r.t. $P$
 7:     $T \leftarrow T \cup \{best\_triplet\}$
 8:     Remove all pairs in $best\_triplet$ from $P$
 9: **end while**
10: $r \leftarrow |T| \bmod 220$
11: **if** $r \ne 0$ **then**
12:     Sample $220 - r$ triplets randomly from $S$ and add to $T$
13: **end if**
14: **return** $T$

---

## C Supplementary Evaluation and Analysis Procedures

### C.1 RSA objective

During training we use Pearson-correlation RSA on z-scored pairwise distances. Pearson is smooth, so gradients propagate from the correlation through distances back to the embeddings. (For evaluation we report Spearman $\rho^2$, which is rank-based and non-differentiable.)

### C.2 Variance partitioning analysis

We model human distances $d_{\text{human}}(i, j)$ with multiple regression using model distances as predictors. For models $X_1, X_2, \ldots$, we fit

$$\hat{d}(i, j) = \beta_0 + \sum_m \beta_m \, d_{X_m}(i, j)$$

over all video pairs in the test split, and report $R^2$. Unique and shared contributions are obtained by comparing nested models (e.g., unique $X_1$ is $R^2_{X_1, X_2} - R^2_{X_2}$); confidence intervals are computed via bootstrap over pairs. We use the best language model as one predictor, and the pretrained and fine-tuned TimeSformer as the other predictors.

### C.3 Split–half reliability

We estimate a noise ceiling for the human RSM with a split–half procedure that respects unequal judgments per pair. In each of 1,000 iterations we: (1) restrict to lower-triangle pairs with at least two ratings; (2) reconstruct binary votes ("similar"/"dissimilar") for each pair using its observed proportion and count, shuffle, and split the votes into two halves; (3) compute the proportion "similar" in each half for every pair and take the Spearman correlation across pairs between halves; (4) average these correlations over iterations and apply the Spearman–Brown correction to estimate full-sample reliability. We report this corrected average as the split–half noise ceiling for the human judgments. In figures, we label this as *split–half* $R^2$, i.e., the squared Spearman–Brown–corrected split–half correlation.

## D Code and data availability

All code used in this paper and our sentence captions are publicly available: (https://drive.google.com/drive/folders/1qoH82510A7Wdgfn_MtdwdWnBEGi9TS6O?dmr=1&ec=wgc-drive-globalnav-goto). The videos shown to participants for the triplet OOO similarity judgments task and therefore are from the Moments in Time (MiT) dataset (http://moments.csail.mit.edu). The MiT license restricts public release of videos from the dataset, and so we ask to please contact the authors for access.

## E Action Recognition Performance

We include here the full results of the UCF101 linear-probe evaluation. All backbone parameters were frozen, and a linear classifier was trained on top of [CLS] features extracted from the pretrained and fine-tuned TimeSformer models. Training was repeated across three random seeds, and Top-1 accuracy is reported as mean $\pm$ standard deviation.

Table 1: Linear probe Top-1 accuracy (%) on UCF101 split1 with frozen backbones. Reported as mean $\pm$ standard deviation over 3 seeds.

| Backbone | Top-1 (%) |
|---|---|
| Pretrained | $95.75 \pm 0.18$ |
| Fine-tuned | $95.70 \pm 0.14$ |

# F    MODEL PERFORMANCE AND SUPERVISION BUDGET

Table 2: Model performance and supervision constraints budget (— indicates not applicable).

| Model UID | Explained Variance ($R^2$) | OOO Accuracy | Constraints/epoch |
|---|---|---|---|
| *Finetuned Models* | | | |
| timesformer-ft-hybrid | 0.162023 | 74.46% | 12978 |
| timesformer-ft-triplet-match | 0.156857 | 66.58% | 12978 |
| clip-vit-b-32-hybrid-finetuned | 0.155953 | 67.84% | — |
| timesformer-ft-triplet | 0.145600 | 70.65% | 12240 |
| videomae-hybrid-finetuned | 0.123535 | 64.56% | — |
| timesformer-ft-rsa | 0.121153 | 63.86% | 13038 |
| *Video Models* | | | |
| x3d-m | 0.123559 | 68.48% | — |
| x3d-s | 0.105202 | 64.67% | — |
| x3d-xs | 0.103721 | 64.95% | — |
| timesformer-base | 0.102408 | 63.59% | — |
| i3d-r50 | 0.094969 | 67.66% | — |
| c2d-r50 | 0.090121 | 65.76% | — |
| slow-r50 | 0.086501 | 67.93% | — |
| slowfast-r50 | 0.085466 | 64.95% | — |
| videomae-base-finetuned-kinetics | 0.086675 | 66.75% | — |
| *Language Models* | | | |
| paraphrase-multilingual-mpnet-base-v2 | 0.134374 | 70.38% | — |
| mxbai-embed-2d-large-v1 | 0.122445 | 66.58% | — |
| paraphrase-multilingual-MiniLM-L12-v2 | 0.120615 | 67.39% | — |
| distiluse-base-multilingual-cased-v1 | 0.110899 | 64.95% | — |
| paraphrase-MiniLM-L6-v2 | 0.102647 | 65.49% | — |
| all-distilroberta-v1 | 0.101303 | 63.04% | — |
| stsb-distilroberta-base-v2 | 0.098953 | 64.13% | — |
| mxbai-embed-large-v1 | 0.090592 | 67.39% | — |
| all-roberta-large-v1 | 0.088598 | 63.04% | — |
| all-mpnet-base-v1 | 0.086371 | 66.58% | — |
| all-mpnet-base-v2 | 0.085562 | 64.67% | — |
| all-MiniLM-L6-v1 | 0.078124 | 65.22% | — |
| all-MiniLM-L6-v2 | 0.077037 | 65.49% | — |
| multi-qa-MiniLM-L6-cos-v1 | 0.068142 | 64.40% | — |
| all-MiniLM-L12-v2 | 0.065997 | 67.39% | — |
| LaBSE | 0.052770 | 61.96% | — |
| clip-ViT-B-32-multilingual-v1 | 0.052506 | 62.77% | — |
| FacebookAI/roberta-base | 0.025612 | 59.24% | — |
| FacebookAI/xlm-roberta-base | 0.022418 | 49.46% | — |
| FacebookAI/roberta-large-mnli | 0.016395 | 47.83% | — |
| FacebookAI/xlm-roberta-large | 0.010090 | 57.07% | — |
| *Image Models* | | | |
| clip-vit-b-32 | 0.126510 | 69.38% | — |
| dino-dino-vitb16 | 0.075451 | 60.77% | — |
| *Vision-Language Models* | | | |
| Qwen3-VL-2B-Instruct | 0.128127 | 68.75% | — |

**Matching Constraints.** Despite the same number of optimizer steps across all approaches, the hybrid objective includes an additional RSA term, introducing a modest number of extra supervision signals ($\approx 738$ pairwise constraints per epoch) beyond the triplet loss (12,240 pairwise constraints). To ensure a fair comparison, we trained a *triplet-only (budget-matched)* variant by adding the same number of extra triplet constraints each epoch. This budget-matched triplet model slightly outperforms standard triplet-only training, confirming that more constraints help. Yet, it still underperforms compared to the hybrid model, indicating that the RSA term contributes qualitatively different information by enforcing global structure beyond what can be achieved by simply adding more triplet comparisons.

Table 3: Subset – Finetuned Models along with best Video, Language, and Image model performance.

| Model UID | Explained Variance ($R^2$) | OOO Accuracy |
|---|---|---|
| *Finetuned* | | |
| timesformer-ft-hybrid | 0.162023 | 74.46% |
| timesformer-ft-triplet-match | 0.156857 | 66.58% |
| clip-vit-b-32-hybrid-finetuned | 0.155953 | 67.84% |
| timesformer-ft-triplet | 0.145600 | 70.65% |
| videomae-hybrid-finetuned | 0.123535 | 64.56% |
| timesformer-ft-rsa | 0.121153 | 63.86% |
| *Baseline Models* | | |
| timesformer-base | 0.102408 | 63.59% |
| clip-vit-b-32 | 0.126510 | 69.38% |
| videomae-base-finetuned-kinetics | 0.086675 | 66.75% |
| *Best Video Model* | | |
| x3d-m | 0.123559 | 68.48% |
| *Best Language Model* | | |
| paraphrase-multilingual-mpnet-base-v2 | 0.134374 | 70.38% |
| *Best Image Model* | | |
| clip-vit-b-32 | 0.126510 | 69.38% |
| *Best Vision-Language Model* | | |
| Qwen3-VL-2B-Instruct | 0.128127 | 68.75% |

# G ADDITIONAL METHODS

## G.1 SENTENCE CAPTIONING OF VIDEOS

Sentence captions were used from a publicly available dataset Garcia et al. (2025). Here we briefly describe the captioning procedures, for full details refer to the original paper. A group of 150 participants was recruited on Prolific to provide sentence captions. Eligibility criteria for this online study required participants to be native English speakers, 18 years or older (M=39.72, SD=13.24), with normal or corrected-to-normal vision. The cohort consisted of 63 females and 87 males. Self-reported race and ethnicity were as follows: 114 white, 14 black, 10 Asian, 9 mixed race, 2 other, and 3 who declined to report.

The task required each participant to write a single-sentence caption for 12 videos presented in a random order (10 standard and 2 catch trials). Participants typed their responses into a text box that initially showed a placeholder prompt: "Description of the actions and interactions of the people in the video in a single sentence..." (Appendix Figure 5).

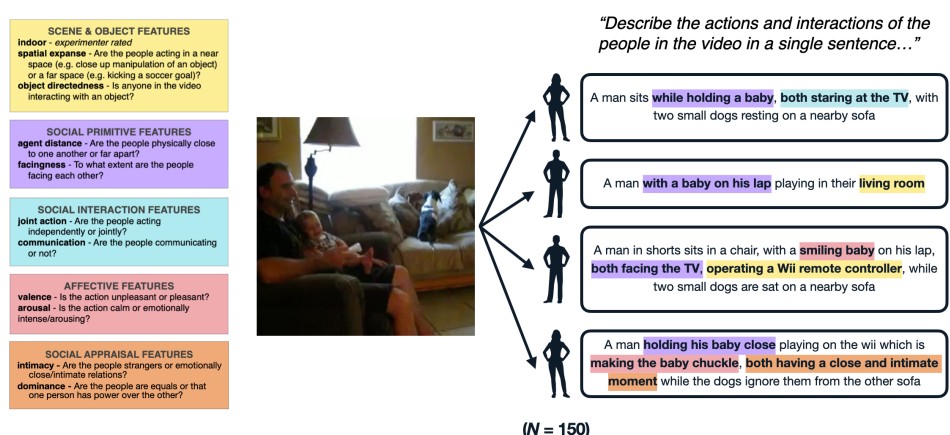

Figure 5: Example video and participant descriptions from the publicly available dataset from Garcia et al. (2025). Participants viewed a short video clip (center) and were asked to describe the actions and interactions of the people in the scene using a single sentence. Example responses (right) show how different aspects of the same video can emphasize distinct feature categories: scene and object features (yellow), social primitive features (purple), social interaction features (blue), affective features (red), and social appraisal features (orange) (Modified from McMahon et al. (2023)). Each highlighted phrase corresponds to the feature dimension it represents.

## G.2 PARTICIPANT DEMOGRAPHICS FOR ODD-ONE-OUT SIMILARITY DATASET.

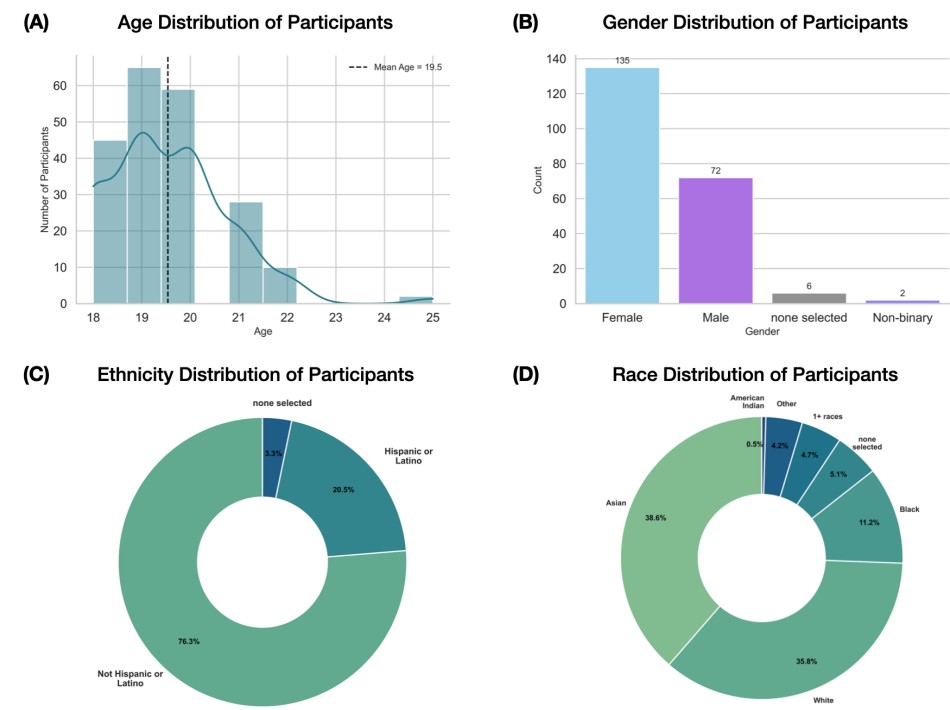

Figure 6: Participant Demographics. (A) Age distribution (mean = 19.5 years). (B) Gender distribution. (C) Ethnicity distribution. (D) Race distribution.

## G.3 VIDEO ACTION CATEGORY DISTRIBUTION

Table 4: Action category frequencies across all 250 videos, showing the full distribution of annotated behaviors from the most common everyday actions to the least frequent, diversity-focused categories.

| category | count | category | count |
|---|---|---|---|
| crying | 21 | building | 2 |
| laughing | 18 | boating | 2 |
| drumming | 18 | baking | 2 |
| brushing | 18 | closing | 2 |
| fishing | 11 | wrestling | 2 |
| dancing | 10 | bowing | 2 |
| giggling | 9 | gardening | 2 |
| cooking | 9 | shopping | 1 |
| discussing | 8 | digging | 1 |
| clapping | 7 | pushing | 1 |
| driving | 6 | unloading | 1 |
| smoking | 6 | exercising | 1 |
| working | 6 | drilling | 1 |
| eating | 6 | applauding | 1 |
| singing | 5 | spitting | 1 |
| planting | 5 | catching | 1 |
| reading | 5 | camping | 1 |
| bathing | 5 | barbecuing | 1 |
| playing | 5 | hugging | 1 |
| kicking | 5 | riding | 1 |
| mowing | 4 | chewing | 1 |
| hiking | 4 | cleaning | 1 |
| throwing | 4 | speaking | 1 |
| skating | 3 | playing+videogames | 1 |
| drinking | 3 | drawing | 1 |
| walking | 3 | skiing | 1 |
| dipping | 3 | jogging | 1 |
| hunting | 3 | studying | 1 |
| knitting | 3 | bowling | 1 |
| | | unpacking | 1 |

## G.4 PROPORTIONS OF NOUNS AND VERBS ACROSS VIDEO CAPTIONS

Table 5: Most frequent nouns and verbs across all captions, showing the proportion of word occurrence across captions.

| Noun | Noun Proportion | Verb | Verb Proportion |
|---|---|---|---|
| man | 0.255 | play | 0.173 |
| baby | 0.203 | sit | 0.083 |
| woman | 0.151 | hold | 0.067 |
| child | 0.133 | look | 0.065 |
| girl | 0.083 | talk | 0.052 |
| boy | 0.081 | cry | 0.051 |
| people | 0.058 | watch | 0.046 |
| drum | 0.052 | stand | 0.046 |
| adult | 0.044 | brush | 0.043 |
| hand | 0.041 | laugh | 0.038 |
| tooth | 0.039 | sing | 0.025 |
| toddler | 0.037 | dance | 0.024 |
| water | 0.032 | appear | 0.024 |
| mother | 0.031 | smile | 0.024 |
| kid | 0.026 | try | 0.022 |
| toy | 0.025 | help | 0.021 |
| fishing | 0.025 | feed | 0.021 |
| person | 0.024 | make | 0.020 |
| car | 0.024 | use | 0.019 |
| lady | 0.022 | lie | 0.019 |
| kitchen | 0.022 | have | 0.019 |
| bed | 0.021 | walk | 0.018 |
| book | 0.019 | fish | 0.018 |
| hair | 0.018 | show | 0.017 |
| father | 0.018 | do | 0.017 |
| dance | 0.017 | seem | 0.017 |
| guy | 0.017 | put | 0.017 |
| food | 0.017 | eat | 0.017 |
| friend | 0.017 | lay | 0.016 |
| front | 0.017 | throw | 0.014 |
| floor | 0.016 | move | 0.013 |
| son | 0.015 | take | 0.013 |
| bath | 0.015 | read | 0.013 |
| arm | 0.015 | face | 0.013 |
| game | 0.015 | wear | 0.012 |
| male | 0.014 | speak | 0.012 |
| brother | 0.014 | explain | 0.012 |
| mouth | 0.014 | get | 0.011 |
| guitar | 0.014 | catch | 0.010 |
| ball | 0.014 | work | 0.009 |
| side | 0.013 | learn | 0.009 |
| dog | 0.013 | prepare | 0.009 |
| playing | 0.012 | smoke | 0.009 |
| room | 0.012 | cook | 0.009 |
| dad | 0.012 | interact | 0.009 |
| ice | 0.011 | discuss | 0.009 |
| microphone | 0.011 | drink | 0.009 |
| boat | 0.011 | clean | 0.009 |
| time | 0.011 | drive | 0.009 |
| face | 0.011 | practice | 0.008 |

# H SUPPLEMENTAL QUALITATIVE INTERPRETABILITY

## H.1 QUALITATIVE EXAMPLES OF HUMAN–MODEL AGREEMENT

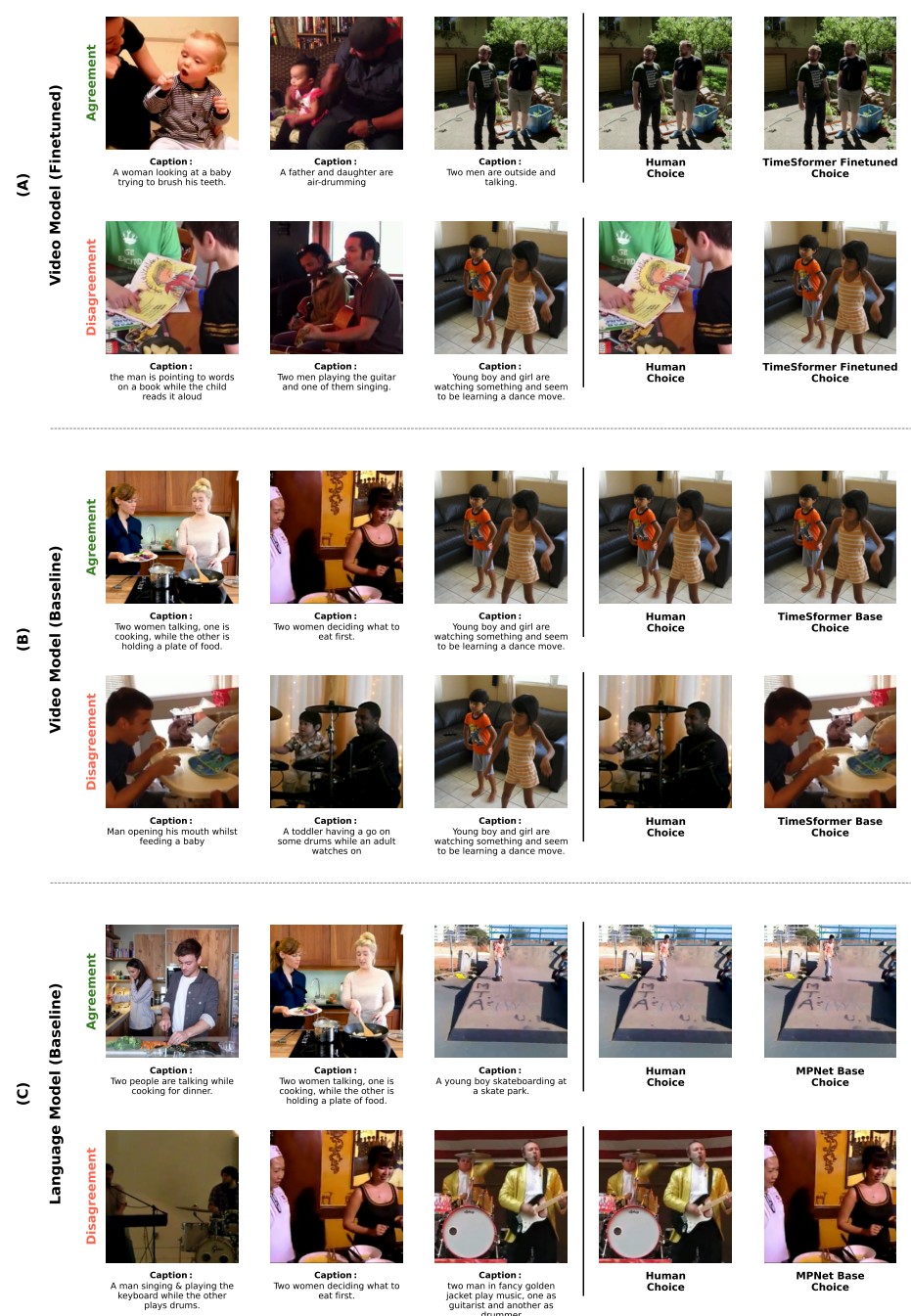

Figure 7: Human–model agreement and disagreement on triplet odd-one-out judgments across three modalities *(A) finetuned TimeSformer*, *(B) baseline TimeSformer*, and *(C) baseline MPNet*, where each row shows the three candidate videos, the human-selected odd one out, the model-selected odd one out, and the human-written captions, with horizontal separators marking modality groups and row labels indicating agreement or disagreement.

## H.2    MODEL-HUMAN AGREEMENT FOR THE SAME SET OF TRIPLETS

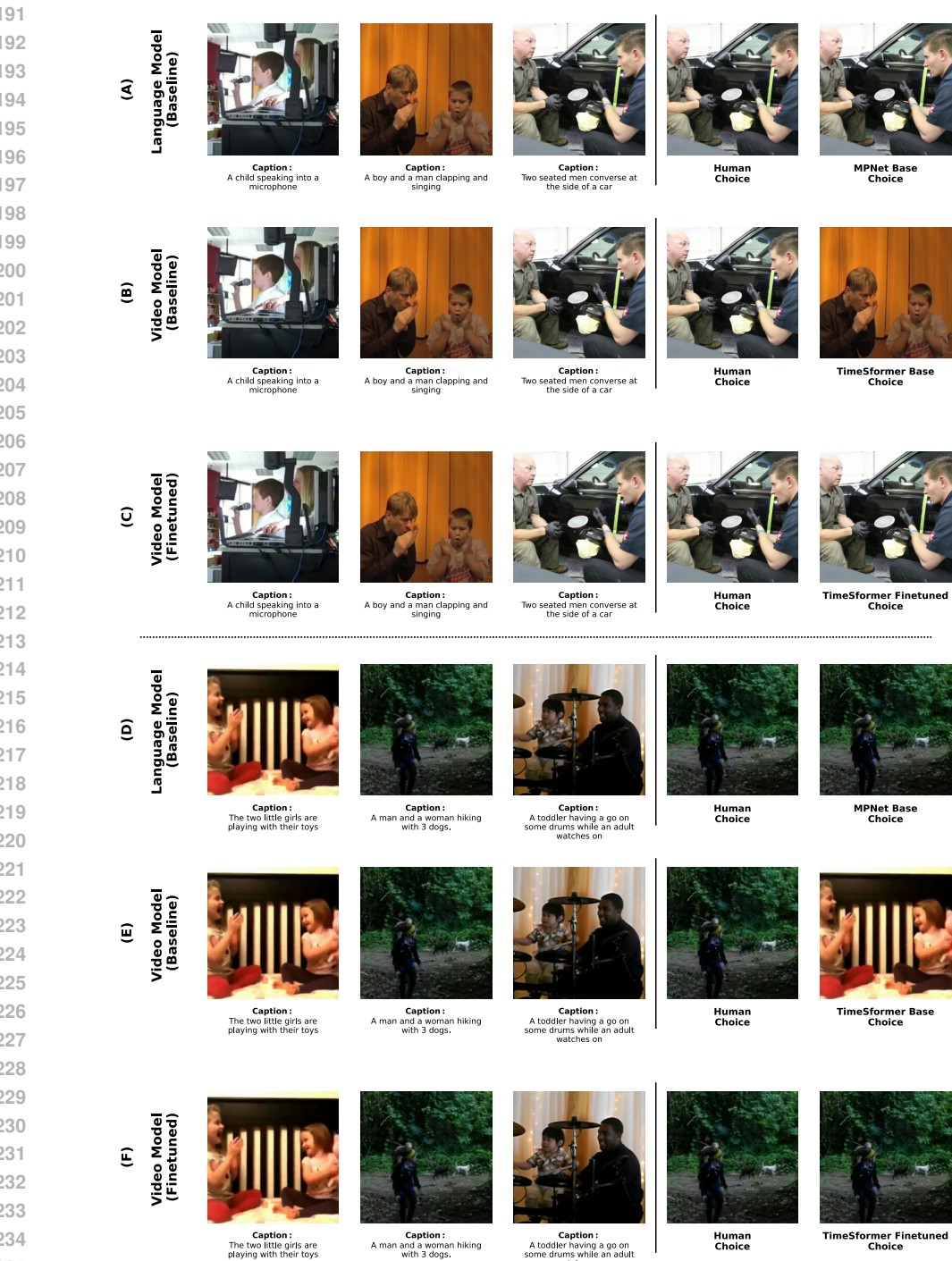

Figure 8: Human–model agreement and disagreement on triplet odd-one-out judgments across three modalities *(A & D) baseline MPNet*, *(B & E) baseline TimeSformer*, and *(C & F) finetuned TimeSformer*, where each row shows the three candidate videos, the human-selected odd one out, the model-selected odd one out, and the human-written captions, with horizontal separators marking modality groups and row labels indicating agreement or disagreement.

## H.3 ATTENTION ROLLOUT COMPARISON

Baseline        Finetuned

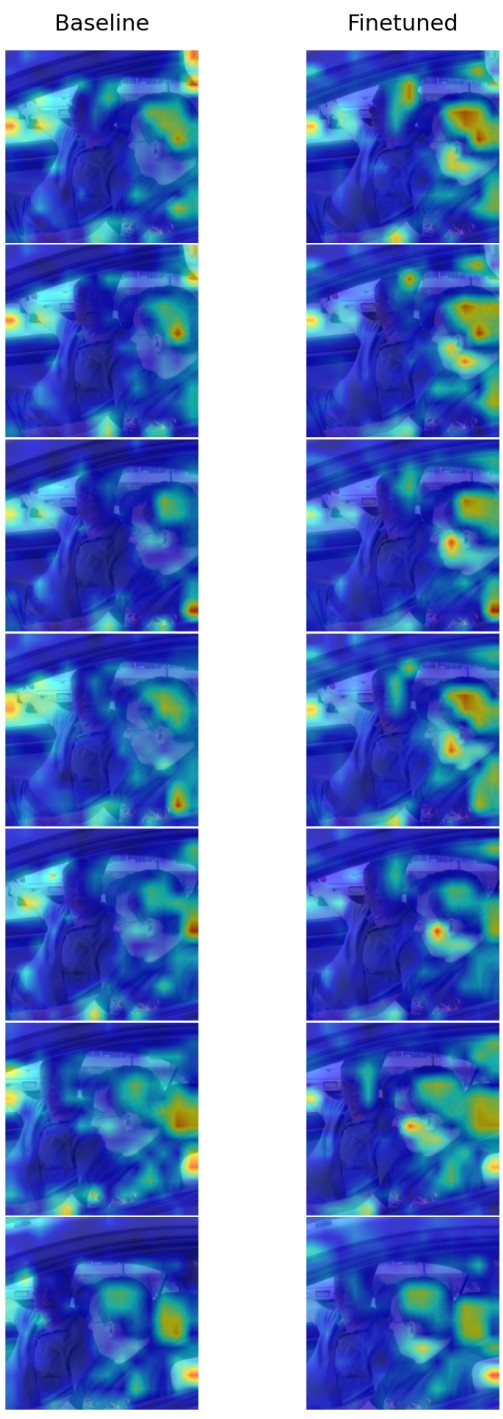

Figure 9: Spatial attention patterns before and after behavior-guided finetuning using attention rollout. We sampled 7 frames from a representative video from the dataset. The pretrained TimeSformer primarily attends to coarse action-relevant regions (e.g., limb motion). In contrast, the finetuned model exhibits more selective attention to social signals: faces, gaze direction, conversational partners, and hands.

