# OpenReview forum: "Aligning Video Models with Human Social Judgments via Behavior-Guided Fine-Tuning"
_ICLR.cc/2026/Conference — Submitted to ICLR 2026_

### Official Review · Reviewer_TRb2 · 2025-10-17

**Soundness:** 3
**Presentation:** 3
**Contribution:** 3
**Rating:** 6
**Confidence:** 4

**Summary:**

This paper presents a compelling investigation into the alignment of video model representations with human social perception. The authors identify a significant and counter-intuitive "modality gap": embeddings from language models applied to video captions are better predictors of human similarity judgments for social videos than embeddings from purpose-built, pretrained video models. To address this misalignment, the work introduces a novel "behavior-guided fine-tuning" methodology. The core of this approach is an innovative hybrid loss function that combines local, triplet-based constraints with a global alignment objective based on Representational Similarity Analysis (RSA), a technique repurposed from an analysis tool into a differentiable training signal.

**Strengths:**

The paper is exceptionally well-motivated. It tackles the timely and critical problem of aligning AI systems with human perception, which is essential for developing more intuitive, interpretable, and trustworthy AI, particularly as these systems are deployed in human-centric environments. The central finding that motivates the work—the "modality gap"—is both surprising and insightful.

The creation and planned release of the social video similarity judgment dataset is a substantial contribution in its own right. Collecting large-scale human behavioral data is a difficult and resource-intensive endeavor, and such datasets are invaluable for the research community. The dataset, with its ~49,000 odd-one-out judgments, provides a rich and dense source of relational supervision that captures the latent structure of human perception far more effectively than discrete categorical labels.

The core technical contribution—the hybrid loss function—is both novel and well-justified. While triplet loss is a well-established method for metric learning, its combination with a differentiable RSA objective is a significant innovation.

The empirical validation of the proposed method is thorough, convincing, and multifaceted. The authors employ a well-designed set of experiments that provide strong support for their claims.

The paper provides nuanced analyses that deepen the reader's understanding. The variance partitioning analysis in Figure 3 is a prime example. It clearly visualizes how fine-tuning not only increases the shared variance between the video and language models but also significantly boosts the unique variance explained by the video model.

The use of Low-Rank Adaptation (LoRA) for fine-tuning is a pragmatic and well-justified choice. By updating less than 2% of the model's parameters, the authors make their approach computationally efficient, scalable, and accessible. This aligns with the broader trend in the field toward parameter-efficient fine-tuning (PEFT) as a sustainable way to adapt large foundation models, making the proposed method highly practical for other researchers to adopt and build upon.

**Weaknesses:**

(W1) The primary concern is the generalizability of the findings, stemming from the limited size of the video stimulus set. While the number of human judgments is large, these judgments are all made on a set of only 250 video clips, which are themselves sourced from a subset of the Moments in Time dataset. This raises the possibility that the model has learned a similarity function that is highly specific to the content, style, and inherent biases of this particular video corpus. The paper acknowledges this limitation but could more deeply consider its implications.

(W2) Furthermore, the paper does not discuss the cultural or demographic properties of either the human annotators or the individuals depicted in the videos. Social perception is known to be culturally situated; what constitutes a "similar" interaction can vary significantly across different cultural contexts. A model aligned with the judgments of one specific annotator population may fail to align with—or even perform poorly for—others. This is a critical consideration under the ICLR Code of Ethics principle to "Be Fair and Take Action not to Discriminate," as there is a risk of encoding a culturally specific viewpoint as a universal ground truth.

(W3) The paper uses TimeSformer, an influential transformer-based video model from 2021, as its experimental backbone. While a reasonable choice, the field of video representation learning has seen significant advances since its publication, particularly in the area of self-supervised learning. The authors themselves note the promise of modern architectures like V-JEPA. The magnitude of the initial "modality gap" could be partially an artifact of the specific limitations of the chosen baseline. It is plausible that a more recent self-supervised model, trained to learn richer and more generalizable representations of the world, might exhibit a smaller initial gap with human judgments. Without testing the fine-tuning method on a more contemporary backbone, it is difficult to know if the reported gains are specific to improving an older architecture or if they represent a more fundamental benefit that would also apply to state-of-the-art models.

(W4) The implementation details of the hybrid loss function could be specified more clearly. The paper states that the weighting parameter $\alpha$ is fixed at 0.7, while $\beta$ is linearly ramped from 0.3 to 0.7 over the course of training.1 However, the rationale for these specific values and the ramping schedule is not provided. It is unclear whether these are the result of a systematic hyperparameter search or an ad-hoc choice.

(W5) The strength of the language model baseline is a cornerstone of the paper's motivation, yet its validity hinges on the quality of the video captions. The paper describes them as "brief descriptive captions" but provides no examples or details on their generation process. If the captions are highly abstract and semantically rich (e.g., "two friends playfully competing" vs. "two people boxing"), they might contain information that is not easily inferred from the visuals alone, giving the language model a significant advantage. Providing a few examples of the captions would be crucial for readers to properly interpret the modality gap and contextualize the performance of all models.

**Questions:**

1. Regarding the Stimulus Set: Could you provide more detail on the diversity of the 250 video clips, perhaps with a breakdown of the types of social interactions represented? Furthermore, could you comment on the demographic background of both the human annotators and the individuals in the videos, and discuss how this might shape the "ground truth" human similarity structure you are aligning to?

2. Regarding the Choice of Backbone: The selection of TimeSformer is understandable. However, how do you anticipate the initial "modality gap" and the relative gains from your fine-tuning method would change if applied to a more recent, self-supervised video model like V-JEPA or VideoMAE?

3. Regarding the Hybrid Loss Formulation: Could you elaborate on the process for selecting the loss weights ($\alpha=0.7$) and the ramping schedule for $\beta$? Were these values determined through a hyperparameter search, and how sensitive is the model's final performance to these specific choices?

4. Regarding the Language Baseline: To help the reader better interpret the "modality gap," would it be possible to provide a few representative examples of the video captions that were used to generate the language model embeddings?

---

> ### Author Response · Authors · 2025-11-18
> **Updated Paper, Individual Review Response(s), and Overall Review Responses**
>
> Thank you reviewer TRb2 for your positive assessment and insightful suggestions. We are encouraged that you found our work to be both highly timely and an important problem!  As noted above we have updated a new version of our manuscript with responses to all of your suggestions.
>
> Please find responses to your individual suggestions below:
>
> ### **1. Cultural Generalization and Stimulus:**
> Thank you for highlighting this important point. We now discuss the cultural context of our experiment and demographic information about our annotators in the updated paper. All our annotators were native or highly proficient English speakers from a University’s Psychological Research Platform. We now report their demographic breakdown in the appendix. In addition the videos and captions we use are from a publicly available US dataset with captions provided from native English speakers. As the reviewer notes, social similarity judgments can vary greatly across cultures. To directly address this concern, we have also added a discussion of these points to Limitations explaining the cultural specificity of our results and stating the fact that our results should not be assumed to be culturally universal. While a full cross-cultural evaluation is beyond our current scope, we discuss the importance of this future direction (last paragraph of Section 5.3: Limitations).
>
> ### **2. Choice of TimeSformer and FollowUp:**
> We appreciate the reviewer’s concern that while TimeSformer is an influential video model, it is slightly outdated. We originally chose TimeSformer because it was the strongest transformer among the pretrained models we initially benchmarked, making it compatible with LoRA fine-tuning. To address your concern, we now include benchmarking of modern image models: CLIP-ViTb32 and DINOv2, and a more modern video model VideoMAE. Interestingly these three more modern models all perform worse than the best language model and the fine-tuned Timesformer (interestingly DINOv2 and VideoMAE are worse than Timesformer before pre-training). To further strengthen our claims, we added additional experiments where we apply our novel fine-tuning methods to CLIP and VideoMAE and find that both are significantly improved, highlighting the generalization of our results. These results are now included in updated Figure 2.  The addition of these more modern models strengthens our claims and underscores overall conclusions of the paper.
>
>
> ### **3. Hybrid Loss:**
> Thank you for asking for clarification and allowing us the opportunity to strengthen the paper! We are happy to clarify these details here and in the revision: We have added more clarification to the paper under the Section 3.3.1 Hybrid Loss Function. We identified this schedule as optimal early in our experiments, and, if time permits, we will add an ablation in the appendix showing that it yields more stable training and validation losses than fixed schedules using alternative values.
>
> ### **4. Language Model Baseline, Caption Quality and Examples:**
> Thank you for this great suggestion! To better illustrate the nature of the captions we used, we included a much more detailed description. The captions were from a prior, published dataset, and we now include (1) an overview of how these captions were collected (Appendix Section G.1), (2) representative captions and still frames for some of the videos in our dataset to show examples of model-human choice agreements and disagreements (Appendix Section H.1), and (3) variations among the type of captions for a specific video (Appendix Section G.1, Figure 5). By including these clarifications and examples, we believe the reader now has a more comprehensive understanding of the language baseline results and context.

---

> > ### Author Response · Authors · 2025-11-26
> > **New paper with additional changes, including dataset details**
> >
> > We are glad the reviewer found our work to be “extremely well motivated” and "insightful”. We believe that, with your suggestions, we have been able to even further improve the manuscript and we hope the reviewers agree. We are eager to hear if there are any ways to further improve the manuscript
> >
> > **Dataset diversity:** We now updated the paper reflecting your final questions on the dataset. We have  added additional information about the video dataset and captions. Videos were 3-second clips of everyday actions sub-sampled from the Moments in Time dataset. The videos were not labeled with categorical "interaction types" but contained a variety of actions. We now include a table with that breakdown from the original paper in Appendix G.3, Table 4. We have also included a table showing the word occurrence in  the caption content in Appendix G4, Table 5.
> >
> > **Please let us know if you have any additional questions. We believe we have addressed all your points, and if you agree, we kindly ask that you consider updating your score.** Thank you!

---

### Official Review · Reviewer_WHgJ · 2025-10-27

**Soundness:** 2
**Presentation:** 3
**Contribution:** 2
**Rating:** 2
**Confidence:** 3

**Summary:**

The paper proposes a novel benchmark of human similarity judgments on short social video clips and a behavior-guided fine-tuning method that aligns video model representations with human social perception. The authors introduce a hybrid loss combining triplet and RSA objectives and evaluate its effectivenes. Results show improved alignment between model and human similarity judgments as well as enhanced representation of social-affective attributes.

**Strengths:**

1. Interesting Motivation: The idea of aligning model representations with human social perception is an important and timely topic, especially as video-based AI systems become increasingly prevalent in social and interactive contexts.
2. New Dataset Contribution: The benchmark of 49k human odd-one-out judgments on social videos provides valuable behavioral data that could support future studies on human-model alignment.
3. Novel Fine-Tuning Approach: The hybrid triplet–RSA loss is methodologically sound and creatively combines local and global representational alignment.
4. Clear Presentation: The paper is clearly written, provides reproducibility details, and the methods are well-documented.

**Weaknesses:**

1. Cultural Diversity Ignored – The work does not address cross-cultural or contextual differences in social judgment. Since human similarity judgments are deeply culture-dependent, the benchmark risks encoding culturally specific biases. This limitation should be discussed in more depth and ideally evaluated empirically.
2. Weak Baselines and Model Choices –
    * The use of TimeSformer pretrained on Kinetics-400 as the main video backbone is a significant limitation. Kinetics-400 is small by current standards and biased toward action recognition, not social understanding.
    * Given the state of the field, stronger and more modern baselines (e.g., Qwen3-VL, InternVL-3.5, SigLIP-v2, DINOv3, CLIP) should be included. Many of these outperform older video models and are standard in current video vision-language research.
    * It is also well known that video encoders underperform image encoders in general visual understanding due to differences in training data and supervision. Testing the best image-based encoders with frame averaging would be an important baseline.
3. Limited Model Generalization – The authors draw conclusions based on experiments using a model (TimeSformer on Kinetics-400) that is not representative of current top-performing video architectures. Thus, the results and claims about “closing the modality gap” are not convincing in the broader context of modern video-LLMs.
4. Lack of Qualitative Examples – The paper provides no qualitative examples or visualizations of the dataset or model outputs. This omission weakens the reader’s understanding of what constitutes “social similarity” and how the model captures it.
5. Section Naming and Style – Minor: Section 3 should be titled “Method” instead of “Methods” for consistency with standard paper structure.
6. Misaligned with Field Progress – The motivation, while conceptually interesting, does not sufficiently engage with the current landscape of multimodal vision-language research, which already investigates human alignment via large-scale preference learning, contrastive learning, and cross-modal supervision. The contribution therefore feels incremental relative to recent progress.

**Questions:**

see above

---

> ### Author Response · Authors · 2025-11-18
> **Updated Paper, Individual Review Response(s), and Overall Review Responses**
>
> Thank you reviewer WHgj for your thoughtful feedback and for highlighting important areas to strengthen this paper. We appreciate your time and feel confident we are able to address your suggestions to clarify and strengthen the work! As noted above we have updated a new version of our manuscript with responses to all of your suggestions.
> Please find responses to your individual suggestions below:
> ### **1. Stronger Baseline & Modern Models:**
> Thank you for this important suggestion. We initially limited our analyses to video and language models (and transformers for LORA fine-tuning), but you are right to point out that many more modern, multimodal models are image-based models. To address your concern, we now include benchmarking of CLIP-ViTb32 and DINOv2, and more modern video model VideoMAE. Interestingly these three more modern models all perform worse than the best language model and the fine-tuned Timesformer (interestingly DINOv2 and VideoMAE are worse than Timesformer before pre-training). To further strengthen our claims, we added additional experiments where we apply our novel fine-tuning methods to CLIP and VideoMAE and find that both are significantly improved, highlighting the generalization of our results. These results are now included in updated Figure 2.  The addition of these more modern models strengthens our claims and underscores overall conclusions of the paper.
> ### **2. Model Generalization & Contribution:**
> We value your suggestion and we are careful not to overgeneralize our conclusions beyond the tested scope, which is to introduce a methodological framework to align vision representations with human social similarity judgments, rather than to benchmark or endorse any specific backbone architecture.  A major contribution of our work, our novel fine-tuning approach, is model-agnostic and can be applied on top of any backbone. This claim is now strengthened with the additional vision models tested above. We now emphasize these points about generalizability and the contribution of our fine-tuning approach more clearly in the Introduction / Contributions.
> ### **3. Alignment w/ Field Progress:**
> Thank you for highlighting these areas. We believe our research is complementary to other efforts in vision-language/multimodal alignment as it optimizes models directly on human similarity from videos, without explicit cross-modal supervision. While we assess the role of language models to understand human semantic similarity, humans make these social semantic judgments directly from visual input. We believe the above changes to integrate more modern multimodal models (and the persisting “gap” of these models) strengthens our findings and brings it more in line with modern AI. Based on your suggestion, we expanded our Related Work to explicitly acknowledge contemporary efforts in aligning models with human preferences. For instance, we referenced recent large-scale preference learning approaches and cross modal training in VLMs, and the specific ways our work is meant to be complementary to these ongoing efforts (last paragraph of Section 2: Related Work).
> ### **4. Cultural Generalization/Bias:**
> Thank you for highlighting this important point. We now discuss the cultural context of our experiment and demographic information about our annotators in the updated paper. All our annotators were native or highly proficient English speakers from a University’s Psychological Research Platform. We now report their demographic breakdown in the appendix. In addition the videos and captions we use are from a publicly available US dataset with captions provided from native English speakers. As the reviewer notes, social similarity judgments can vary greatly across cultures (Pang et al, 2024). To directly address this concern, we have also added a discussion of these points to Limitations explaining the cultural specificity of our results and stating the fact that our results should not be assumed to be culturally universal. While a full cross-cultural evaluation is beyond our current scope, we discuss the importance of this future direction (last paragraph Section 5.3: Limitations).
> ### **5. Qualitative & Visual Examples:**
> Thank you for this interesting suggestion! We now include in Appendix Fig 7-8 qualitative examples of the dataset and model outputs where the pre-trained language models match humans but vision models do not. Pretrained video models emphasize low-level visual differences while LMs, like humans, are more tuned to social similarity between people. After fine-tuning, video models align with humans. We believe these illustrative examples greatly enhance the paper.
> ### **6. Section Naming:**
> Thank you for this suggestion, we updated the section title to ‘Method’.
>
> Pang, H. T., Zhou, X., & Chu, M. (2024). Cross-cultural Differences in Using Nonverbal Behaviors to
> Identify Indirect Replies. Journal of Nonverbal Behavior, 48(2), 323-344. https://doi.org/10.1007/s10919-024-00454-z

---

> > ### Comment · Reviewer_WHgJ · 2025-11-25
> >
> > Thank you to the authors for their response. While the authors have included additional models such as DINOv2, VideoMAE, and CLIP, none of these represent the current state of the art in their respective domains. It is also important to include evaluations of more recent vision-language models (VLMs), such as those from the Qwen family.
> >
> > Furthermore, reviewer TRb2 requested statistics on the dataset, specifically: “Could you provide more detail on the diversity of the 250 video clips, perhaps with a breakdown of the types of social interactions represented?” Hhowever, I could not find this information updated in the revised version of the paper.
> >
> > Additionally, It would also be valuable to include an analysis of the linguistic characteristics of the dataset, such as the most common nouns and verbs, or semantic similarities between key terms (e.g., “mother” and “father” might be more similar than “mother” and “two men”). Such analyses would help to better understand the underlying data and shed light on why language modelsare better in capturing certain aspects.
> >
> > I would be willing to increase my score if the authors include a more detailed dataset statistics and analysis and evaluate state-of-the-art models such as Qwen3-VL, InternVL-3.5, SigLIP-v2, and DINOv3.

---

> > > ### Author Response · Authors · 2025-11-26
> > > **New paper with dataset characterizations and notes on newer models**
> > >
> > > Thank you for your encouragement and highlighting these additional points for improvement.
> > >
> > > **Dataset diversity.** We have now added additional information about the video dataset and captions. Videos were 3-second clips of everyday actions sub-sampled from the Moments in Time dataset. The videos were not labeled with categorical "interaction types" but contained a variety of actions. We now include a table with that breakdown from the original paper in Appendix G.3, Table 4 as well as the caption content in Appendix G.4, Table 5.
> > >
> > > **Additional, multimodal models.** We would like to note that CLIP while perhaps not the most state-of-the-art, was included in the list you provided us with your original review and is a common multimodal benchmark. We believe its addition provides compelling additional evidence in support of our methods and findings. In addition, we would also like to highlight that newer, more modern models often fail to add meaningful improvements on human-alignment on most visual tasks (Garcia et al., 2025 ICLR; Linsley et al. 2023 NeurIPS; Schrimpf et al., 2018 bioRxiv).
> > > We would also like to stress again that the major contribution of this work is the benchmark and fine-tuning approach, which can be generalized to any model. We have added a discussion of these points to the paper introduction lines 85-86.
> > >
> > > Importantly, newer architectures come with a large compute cost, often beyond our ability to run with compute at the academic level. Nevertheless, given your feedback, we are currently working on adding more modern models suggested, time allowing.
> > >
> > > **In the meantime, we believe that we have addressed all your other points and if so, we would kindly ask that you consider updating your score.**

---

> ### Author Response · Authors · 2025-11-30
>
> *We have now implemented multimodal embeddings from qwen-v3.* We show that the combined vision and language embeddings from this SOTA model still fall short of the best language model and most fine-tuned vision transformers. These results are summarized in Figure 2 and detailed in methods/results. **We believe the addition of this more modern model significantly strengthens our findings and helps the paper make contact with modern computer vision.**
>
> We are confident we have now addressed all the weaknesses and questions in your initial review and responses. We believe the paper is significantly improved thanks to your feedback, and are grateful for your engagement and constructive feedback. Thank you again!

---

### Official Review · Reviewer_BVfZ · 2025-10-31

**Soundness:** 3
**Presentation:** 3
**Contribution:** 3
**Rating:** 4
**Confidence:** 3

**Summary:**

The authors introduced a new dataset of approximately 49,000 human "odd-one-out" similarity judgments on 250 short social video clips. Using the similarity dataset, the authors demonstrated that embeddings from caption-based language models align better with human visual judgments than pretrained video models. Furthermore, the authors fine-tuned a TimeSformer video model with a novel hybrid triplet-RSA loss. This new model achieves superior human alignment and demonstrates an improved ability to encode social-affective attributes.

**Strengths:**

- this is a valuable and timely investigation into the alignment of video model representations with human social perception, especially on action understanding / social judgment tasks
- the identified modality gap is interesting.  It clearly motivates the need for new approaches to align video models with human-like representations
- the behavior-guided fine-tuning improves the encoding of social-affective attributes, which is a surprising result

**Weaknesses:**

While the results seem interesting, I have a few concerns:
- Depending on the nature of video captions, the modality gap of similarity alignment may or may not be that interesting. Generally, captions capture information at a higher semantic level than visual features, which may be why language models align well with human judgment. For example, language embeddings for captions like “basketball” and “tennis” might be reasonably close since they are both ball sports, but the visual scene for these two sports are vastly different. So it is almost expected that the language embedding performs closer to human behavior, as the vision task could be more difficult due to the much more noisy information.
- While the authors evaluated an extensive list of models , they are generally pre-trained on smaller datasets e.g., Kinetics-400. There lacks an evaluation of large-scale foundational vision models like CLIP and DINO.
- While not a decisive weakness as it's not really within the scope of this paper, the action recognition accuracy did not improve after behavior-guided fine-tuning . This could imply limited generalizability of the fine-tuned model to other, broader visual tasks.

**Questions:**

- Could the authors provide qualitative examples on dataset triplets? In particular, it would be illustrative to see triplets where language embeddings are aligned with human judgments but the video embeddings are not.
- Could the authors evaluate larger-scale foundational vision models like CLIP and DINO? A CLIP-style model would be particularly interesting since it explicitly enforces vision-language alignment. In these models, is there still the "modality gap" on similarity alignment?

---

> ### Author Response · Authors · 2025-11-18
> **Updated Paper, Individual Review Response(s), and Overall Review Responses**
>
> Thank you for your thoughtful assessment and suggestions. We are encouraged that you found our work to be both highly contributing and interesting! We are also encouraged that you found our problem important. As noted above we have updated a new version of our manuscript with responses to all of your suggestions.
>
> Please find responses to your individual suggestions below:
>
>
> ### **1. Significance of the Modality Gap:**
> We appreciate the point that the superior alignment of caption based language embeddings might be expected given the presence higher lever semantic information. Indeed recent work has shown that text descriptions can predict human similarity judgements better than image/video/audio models in many cases (Marjieh et al., ICLR 2023). However, our findings (1) underscore that this information is automatically extracted from humans in visual tasks, and (2) show it is nevertheless missing from pre-trained vision models. Our contribution, therefore, is to demonstrate this gap in a new dataset and setting (social video clips) and then close it via finetuning. Notably, after our behavior guided finetuning, the TimeSformer’s alignment (and other vision models, see below) with humans surpasses the language model’s.
>
>
> ### **2. Stronger Baseline and Modern Models:**
> Thank you for this important suggestion. We initially limited our analyses to video and language models (and transformers for LORA fine-tuning), but you are right to point out that many more modern vision transformers are available, including multimodal models. To address your concern, we now include benchmarking of two image-based vision transformers CLIP-ViTb32 and DINOv2, and a more modern video model VideoMAE. Interestingly these three more modern models all perform **worse than the best language model and the fine-tuned Timesformer** (interestingly DINOv2 and VideoMAE are also worse than Timesformer before pre-training). To further strengthen our claims, we applied our novel fine-tuning methods to CLIP and VideoMAE and found that both are significantly improved beyond the pre-trained baseline, highlighting the generalization of our results. These results are now included in updated Figure 2.  The addition of these more modern models **strengthens our claims and underscores overall conclusions of the paper.**
>
> ### **3. Qualitative examples:**
> Thank you for this great suggestion! We now include in Appendix figure 8 shows examples where the pre-trained language models match humans but vision models do not. Pretrained video models emphasize low-level visual differences while LMs, like humans, are more tuned to social similarity between people. After fine-tuning, video models align with humans. We believe these add a new dimension to the paper to help readers understand the qualitative factors humans use to judge the videos and the qualitative differences between models.
>
>
> ### **4. Behavior Fine-Tuning vs Action Recognition:**
> Thank you for this thoughtful observation. To clarify, the pretrained TimeSformer we begin with already achieves a strong SOTA accuracy on the action-recognition task. Separate work has also shown that these action recognition categories often do not align with human similarity structure (Dima et al., eLife 2022). After behavior-guided fine-tuning, the model reaches effectively unchanged performance. Our goal in reporting this was not to improve action-recognition accuracy, but to demonstrate that socially guided fine-tuning does not induce catastrophic forgetting on core visual tasks. We have clarified this in the revised version (See Section 3.3.3)
>
>
>
> Marjieh, R., Van Rijn, P., Sucholutsky, I., Sumers, T., Lee, H., Griffiths, T. L., & Jacoby, N. (2023, February 1). Words are all you need? Language as an approximation for human similarity judgments. Poster presented at ICLR 2023. https://openreview.net/forum?id=O-G91-4cMdv

---

> > ### Comment · Reviewer_BVfZ · 2025-11-22
> > **Thank you for the response**
> >
> > I thank the authors for their detailed response. From the qualitative examples, it does seem like the video titles are at a higher semantic level than the video content, making it an "easier" task in that regard, and explains the modality gap. However, I agree with the authors that the fact that vision models do not align with human at this semantic level exposes meaningful opportunity worth investigating.
> >
> > I also appreciate the additional benchmarking results. It is interesting that even CLIP-style models perform worse than the best language model. Could the authors also provide information on what's the input to CLIP and DINO? Are the authors inputting only one frame, or sampling multiple frames?
> >
> > With the additional details provided, I'm happy to raise my score. I do think that there are some opportunities to improve the paper / as future work: while the fine-tuning results look promising, it's not yet clear whether they actually improves downstream tasks. As an example, it would be interesting to see if behavior guided fine-tuning affects how the model see humans in videos--e.g., using GradCAM style visualizations on selected frames, is there a difference between what the model sees pre vs post fine-tuning? Just a thought :)

---

> > > ### Author Response · Authors · 2025-11-25
> > >
> > > Thank you for your support and additional questions/suggestions.
> > >
> > > For image models, CLIP and DINO we extract 7 evenly spaced frames from the video (selected to approximately match the average frame rate of most video models), and average their embeddings. This is detailed in the updated Methods lines 223-225. Pilot studies found very similar results when embeddings were averaged or concatenated, so we averaged for computational efficiency.
> > >
> > > The frame-wise visualizations for fine-tuned versus pre-trained response is an interesting suggestion! We are looking into GradCam for this and hope to incorporate these results if time allows.

---

> > > > ### Author Response · Authors · 2025-11-26
> > > > **Thank you and new revision with visualizations**
> > > >
> > > > Thank you again for your insightful suggestion and for raising your score. We agree that understanding how finetuning changes what the model “pays attention to” is important. We initially explored Grad-CAM; however, it is not well suited for TimeSformer. After the patch embeddings are flattened and mixed by multi-head attention, the hidden states no longer correspond to a fixed spatial grid, so Grad-CAM tends to produce uniformly blue or uninterpretable maps, especially after our global finetuning objective.
> > > >
> > > > Instead we added an attention rollout comparison visualization for vision transformers to both the pretrained and finetuned models in the Appendix (see H.3, Figure 9). The visualizations show that the pretrained model attends mostly to scene context, whereas the finetuned model focuses more on socially informative signals such as face, gaze, and interactions, providing qualitative evidence that the finetuning does change how the model interprets humans in videos and further strengthening our claims.
> > > >
> > > > We are glad the reviewer found our work to "achieve superior human alignment" and "an improved ability to encode social-affective attributes". We believe that, with your suggestions, we have been able to even further improve the manuscript (and make it more interesting!) and we hope the reviewers agree.
> > > >
> > > > We are eager to hear if there are any ways to further improve the manuscript or if you feel your revisions have been fully addressed.

---

> > > > > ### Comment · Reviewer_BVfZ · 2025-11-27
> > > > > **Thank you for the response**
> > > > >
> > > > > I thank the authors for their effort on the new revision. The attention rollout visualization is very helpful -- it does seem like the finetuned model is more attentive to humans rather than to the background scenes. I'm happy to raise my score. Looking forward to future work on investigating this behavior / integrating the findings with large-scale vision language models :)

---

### Author Response · Authors · 2025-11-18
**General Review Response**

We are grateful to the reviewers for their overall positive and constructive feedback.We have now made several major changes based on their feedback that we outline below. All changes are shown in blue text in the revised manuscript.

We believe the manuscript is significantly improved thanks to this feedback and we hope the reviewers agree. We are eager to hear if the reviewers believe there are remaining areas where we can still improve the manuscript.


### **1. Generalizability to more modern vision models.**
Based on the suggestions from all reviewers we have included more modern vision models to our set, including image models with and without language-alignment (DINO-v2, CLIP-ViTb32) and a more modern video model that was trained on a self-supervised task, rather than Kinetics (VideoMAE). Our major claims of the (1) modality gap and (2) fine-tuning still hold (updated Figure 2). All new models are worse than the pre-trained language models, and we further show that both CLIP and VideoMAE can be improved with LORA and our novel dual loss in the same way as Timesformer. These findings allow the paper to make more contact with modern computer vision and significantly strengthen our overall claims and conclusions.

### **2. Clarified impact and contributions of the paper.**
We clarified the importance of the modality gap, which demonstrates (1) that humans spontaneously use semantic structure (captured by LM embeddings of captions) when judging video similarity, pre-trained vision models (now including more modern models) perform worse than the best language model. To further strengthen our claims, we added additional experiments where we apply our novel fine-tuning methods to CLIP and VideoMAE and find that both are significantly improved, highlighting the generalization of our results. These results are now included in updated Figure 2. The addition of these more modern models strengthens our claims and underscores overall conclusions of the paper. Our major contribution is a novel dataset and loss function to fine-tune vision models to close this gap, which we now clarify in Introduction / Contributions. The addition of additional vision transformers (point 1 above) significantly strengthens both these claims and we believe the paper is significantly more impactful thanks to these suggestions.

### **3. Added qualitative examples and captions.**
We have added qualitative examples of triplet judgments from different AI models (language models, and vision with and without fine-tuning) compared to humans (Appendix H, Figure 7-8). These examples compare the OOO choices made between a representative human participant and (A) a finetuned video model, (B) a baseline/pretrained video model, and (C) a baseline language model. These illustrative examples show how a pre-trained video model may emphasize visual similarity over the social structures that humans use. In contrast, both the language models and fine-tuned video models are able to capture this human-like structure. We have also added  more context on captions we have for the videos, including examples across videos and raters (Appendix Figure 5).  We believe these add a new dimension to the paper to help readers understand the qualitative factors humans use to judge the videos and the qualitative differences between models.

### **4. Cultural specificity and subject demographics.**
WHgJ and TRb2 raised the important concern about the cultural specificity of our results. We are grateful that they raised this point, and we now report the demographic breakdown of our subjects in Appendix G.2. We also acknowledge that the publicly available dataset we used was collected from a US dataset (Moments in Time) with native English speaking subjects providing captions. To directly address this concern, we have also added a discussion of these points to Limitations (last paragraph, Section 5.3) explaining the cultural specificity of our results. While a full cross-cultural evaluation is beyond our current scope, we discuss the importance of this as an avenue for future directions.

---

> ### Author Response · Authors · 2025-11-26
> **New revision with additional edits**
>
> We thank the reviewers for their helpful feedback. Based on this we have made additional changes to the manuscript.
>
> Dataset characterization. We have added additional characterization of the dataset to provide information on the action categories, and most common nouns and verbs in the captions (Appendix G3-G4, Tables 4-5). We believe this additional context helps highlight the diversity and focus of the stimuli used in our behavioral task and for model fine-tuning.
> We added additional visualizations to highlight the qualitative changes to the vision models after fine-tuning (Appendix H.3, Figure 9). This illustrative example shows how behavioral fine-tuning changes the “focus” of the model away from background towards faces (particularly gaze) in the videos.
>
> Highlighted the overall contribution of our work. We believe the addition of CLIP and DINO in the previous revision significantly strengthen our findings. New, large, multimodal models are constantly being added, and these are not always feasible to run quickly on academic computing clusters (though we are trying!) However, we think it is unlikely newer more modern models would change the findings of our results. We now discuss prior work showing that task optimization / increasing tunable parameters does not always improve a model’s human alignment (lines 85-86), and emphasize that our major contributions are the human similarity dataset and fine-tuning methods, which can be applied to any vision transformer.
>
> Additional specific points are outlined in response to each reviewer. Overall we believe the paper is significantly improved and are grateful for the reviewers positive and constructive feedback.

---

### Author Response · Authors · 2025-11-30
**General Review Response [3rd]**

We thank the reviewers for their comments and suggestions. We have updated a final version of our paper that addresses all comments and questions from each reviewer. Detailed response are written in response to each reviewer. Below we summarize the major revisions to the paper, so they are available to the reviewers and new AC in a single post.

We believe these additions have significantly strengthened the paper, and are grateful to the reviewers for their useful feedback and encouragement.

### **1. Generalizability to more modern vision models.**
Based on the suggestions from all reviewers we have included more modern vision models to our set, including image models with and without language-alignment (DINO-v2, CLIP-ViTb32), a more modern video model that was trained on a self-supervised task, rather than Kinetics (VideoMAE), and multimodal embeddings (vision and language) from a modern VLM (qwen-v3). Our major claims of the (1) modality gap and (2) fine-tuning still hold (updated Fig 2). All new models are worse than the best pre-trained language model, and we further show that both CLIP and VideoMAE can be improved with LORA and our novel dual loss in the same way as Timesformer. These findings allow the paper to make more contact with modern computer vision and significantly strengthen our overall claims and conclusions.

### **2. Clarified paper impact and contributions.**
We clarified the importance of the modality gap, which demonstrates (1) that humans spontaneously use semantic structure (captured by LM embeddings of captions) when judging video similarity, pre-trained vision models (now including more modern models) perform worse than the best language model. To further strengthen our claims, we added additional experiments where we apply our novel fine-tuning methods to CLIP and VideoMAE and find that both are significantly improved, highlighting the generalization of our results. These results are now included in updated Fig 2. The addition of these more modern models strengthens our claims and underscores overall conclusions of the paper.

Our major contribution is a novel dataset and loss function to fine-tune vision models to close this gap, which we now clarify in Introduction / Contributions. The addition of additional vision transformers (point 1 above) significantly strengthens both these claims and we believe the paper is significantly more impactful thanks to these suggestions.

### **3. Added qualitative examples & captions.**
We have added qualitative examples of triplet judgments from different AI models (language models, and vision with and without fine-tuning) compared to humans (Appendix H, Fig 7-8). These examples compare the OOO choices made between a representative human participant and (A) a finetuned video model, (B) a baseline/pretrained video model, and (C) a baseline language model. These illustrative examples show how a pre-trained video model may emphasize visual similarity over the social structures that humans use. In contrast, both the language models and fine-tuned video models are able to capture this human-like structure. We have also added  more context on captions we have for the videos, including examples across videos and raters (Appendix Fig 5).  We believe these add a new dimension to the paper to help readers understand the qualitative factors humans use to judge the videos and the qualitative differences between models.

### **4. Attention patterns before & after finetuning.**
Based on feedback from BVfZ, we used feature visualization to understand how finetuning changes what the vision model “pays attention to”. To do this, we added an attention rollout comparison visualization for vision transformers to both the pretrained and finetuned models in the Appendix (see H.3, Figure 9). The visualizations show that the pretrained model attends mostly to scene context, whereas the finetuned model focuses more on socially informative signals such as face, gaze, and interactions, providing qualitative evidence that the finetuning does change how the model interprets humans in videos and further strengthening our claims.

### **5. Dataset details.**
In response to WHgJ and TRb2, we have now added additional information about the video dataset and captions. Videos were 3-second clips of everyday actions sub-sampled from the Moments in Time dataset. The videos contained a variety of actions, and a table with that breakdown from the original paper in Appendix G.3, Table 4 as well as the caption content in Appendix G.4, Table 5. We believe these additional details help contextualize our novel dataset and findings.

---

> ### Author Response · Authors · 2025-11-30
>
> ### **6. Cultural specificity & subject demographics.**
> WHgJ and TRb2 raised the important concern about the cultural specificity of our results. We are grateful that they raised this point, and we now report the demographic breakdown of our subjects in Appendix G.2. We also acknowledge that the publicly available dataset we used was collected from a US dataset (Moments in Time) with native English speaking subjects providing captions. To directly address this concern, we have also added a discussion of these points to Limitations (last paragraph, Section 5.3) explaining the cultural specificity of our results. While a full cross-cultural evaluation is beyond our current scope, we discuss the importance of this as an avenue for future directions.

---

### Meta-Review · Area_Chair_3YRk · 2026-01-07

**Summary:**

This submission was reviewed by three expert reviewers, with the ratings of: borderline reject, reject, and borderline accept. The major concerns from the reviewers are about the modality gap, lack of analysis on larger-scale foundation models, limited generalizability, missing discussions of the cultural and demographic diversity, baselines and model choices, limited contributions w.r.t. the recent literature, the outdated backbone and the corresponding conclusion derived. The authors provided a rebuttal for the concerns, and two reviewers engaged in discussions with the authors.

After carefully going through all the review comments, the authors' rebuttal, and the discussions, it can be seen that some of the concerns and questions raised are addressed by the rebuttal fromthe authors. However, there are still outstanding major concerns not well addressed. There is no strong support for a clear acceptance. As a result, it is unfortunate that the paper in its current form is not ready for publication at ICLR. However, the findings in this paper could be interesting to the community, and the authors are encouraged to further improve their paper by incorporating the review comments for a future submission.

**Reviewer Concerns:**

Concerns that the AC thinks were addressed by the rebuttal: lacks an evaluation of large-scale foundation models; the bahavior-guided fine-tuning on action recognition; the cultural and demographic diversity concern was partially addressed; lack of qualitative examples and examples of the captions; and other minor issues.

Concerns that are still outstanding: the domain gap concern; the cultural/demographic issues still exist, though an explanation was provided; lack of evaluation on recent VLMs; generalization; and implementation details.

**Reviewer Scores:**

According to the review comments, and the rebuttal, for each review the reviewer might have changed their score in the way below, if they had been able to participate fully in the discussion:
* Reviewer BVfZ: borderline reject to borderline accept, or unchanged
* Reviewer WHgJ: reject, unchanged
* Reviewer TRb2: borderline accept unchanged, or to borderline reject.

---

### Decision · Program_Chairs · 2026-01-26

Reject